# An unusual way to validate regional chemistry-transport models

**Laurent MENUT**[1], **Sylvain MAILLER**[1], **Bertrand BESSAGNET**[2], **Guillaume SIOUR**[3], **Augustin COLETTE**[2], **Florian COUVIDAT**[2], **and Frédérik MELEUX**[2]

[1]Laboratoire de Météorologie Dynamique, Ecole Polytechnique, IPSL Research University, Ecole Normale Supérieure, Université Paris-Saclay, Sorbonne Universités, UPMC Univ Paris 06, CNRS, Route de Saclay, 91128 Palaiseau, France
[2]INERIS, National Institute for Industrial Environment and Risks, Parc Technologique ALATA, F-60550 Verneuil-en-10 Halatte, France
[3]Laboratoire Inter-Universitaire des Systèmes Atmosphériques, UMR CNRS 7583, Université Paris Est Créteil et Université Paris Diderot, Institut Pierre Simon Laplace, Créteil, France

*Correspondence to:* Laurent Menut, menut@lmd.polytechnique.fr

**Abstract.** A simple and complementary model evaluation technique for regional chemistry-transport is discussed. The methodology is based on the concept that we can learn more on models performances by comparing the simulation results with observational data available for other time periods than the period originally targeted. First, the usual scores selected in this study (spatial and temporal correlations) are computed for a given period, using co-localised observation and simulation data in time and space. Second, the same scores are calculated for several other years by conserving only the spatial locations and Julian days of the year. The difference between the two score provides complementary insights to the following questions: (i) is the model performing well only because the situation is recurrent? (ii) is the model representative enough of the measurements for all variables? (iii) if the pollutants concentrations are not well modelled, is it due to meteorology or chemistry? In order to synthesise the large amount of results, a new indicator is proposed: the "multi-year variability", designed to compare the several error statistics between all the years of validation and to quantify if the studied period was fairly modelled for the good reasons.

## 1 Introduction

Chemistry transport models (CTM) aim at simulating the air pollutants concentrations in the lowest layers of the atmosphere where humans and the environment can be affected by air pollution. Air pollution results from the presence of chemical components emitted into the atmosphere due to anthropogenic activities and natural sources (biogenic emissions from vegetation, soil erosion, sea salts, volcanic activity, and wild-land fires). CTMs are used to represent the dynamic and chemical processes that drive spatial and temporal features of the atmospheric composition.

To estimate the quality of CTMs, model output results are usually compared with available observations. These comparisons are performed since the models exist: this is crucial to quantify the ability of models to reproduce particular observed events or a general behaviour. Depending on the model resolution and domain size, the comparison between model outputs and observations data may be tricky due to the spatial representativeness of the monitoring stations (Valari and Menut, 2008; Solazzo and Galmarini, 2015). All modelling studies takes into account this problem of model representativeness and, for many years, comparisons between observations and models outputs were performed using complex statistical approaches. A non exhaustive list of validation studies are provided hereafter, Baldridge and Cox (1986) and Cox and Tikvart (1990) proposed the use of error statistics like correlation, bias, Root Mean Squared Error in the specific framework of air quality, *i.e.* the atmospheric composition when criteria pollutant concentrations exceed predefined limit values. Chang and Hanna (2004) also proposed an evaluation framework dedicated to air quality model performance and explained there is not "*a single best evaluation methodology*" and how important it is to use as much as possible evaluation criteria to really well understand model results.

Dedicated tools to model evaluation have been developed such as Appel et al. (2011) and Galmarini et al. (2012), to ensure the use of systematic procedures in the evaluation pro-

cess. In parallel, some studies were dedicated to revisit the way to evaluate models such as Thunis et al. (2012), dedicated to air quality in a policy framework. In this study, they proposed the "Target diagram" to have on the same plot the bias and the RMSE. Complementary to the definition of performance scores to be used, Simon et al. (2012) use these scores to compile photochemical models performances over a large set of data over several years of simulation. This kind of evaluation may also be done in dedicated projects such as the recent AQMEII (Air Quality Model Evaluation International Initiative), comparing chemistry-transport models running both in Europe and Northern America, Vautard et al. (2012); Campbell et al. (2015) or the EURODELTA project, Bessagnet et al. (2016) and in the EMEP (European Monitoring and Evaluation Programme) context in the frame of the United Nation Convention on Long-range Transboundary Air Pollution, Prank et al. (2016). Using comparisons between observations and models outputs, some studies proposed methodologies to decompose the statistical scores in order to estimate the main source of errors, Solazzo and Galmarini (2016). Finally, other studies also use observations to adjust the result by implementing methods to unbias simulation without changing the model, as in Porter et al. (2015) for ozone over the United States.

A fundamental difference between observations data and models results is the coherence of the spatial representativeness of the monitoring stations compared to the model cell (Valari and Menut, 2008; Solazzo and Galmarini, 2015). To quantify the model errors due to mis-representation of physics and chemistry from those only due to representativeness, several methodologies have been developed. These methods are effective but often required important computation time. Among these approaches, ensemble modelling is used in analysis of case studies and forecasting, (Kioutsioukis and Galmarini, 2014; Marécal et al., 2015; Lemaire et al., 2016). By performing several perturbed simulations, a general tendency on the error can be identified. But if the case study consists of a complex real situation, the analysis can be challenging. Adjoint modelling allows tracking the behaviour of chemical species with respect to model input parameters. But it requires tedious model developments and the result is generally valid for an infinitesimal perturbation since the problem to solve was linearized, (Menut, 2003; Pison et al., 2007). In practice, the validity of this approach is limited to chemical species with a long lifetime as presented in Kopacz et al. (2010); Mao et al. (2015). Finally, the common point of all these studies is that they are always using the observations corresponding in time and location to the model cell.

In the present study, a simple method is developed to improve the evaluation of models and to identify the processes responsible for discrepancies of models outputs *versus* observations. In areas where the monitoring network are dense enough, like in Europe, comparisons are performed with observations from surface stations that provide hourly

$O_3$, $NO_2$ concentrations for gases and $PM_{2.5}$ and $PM_{10}$ for particles. Complementary to surface concentrations data, the meteorology is evaluated using meteorological networks providing 2m temperature, 10m wind speed and precipitation rates. In order to quantify the transport of aerosols in dense plumes aloft, observations from lidar or from the AERONET (AErosol RObotic NETwork) program for the optical depth are increasingly used to assess regional models.

For all these variables, temporal and spatial correlations are computed to identify the model capacity compared to observations. First, the correlations are calculated between observations data and model outputs for the simulation year (*i.e.* the reference year). Second, the correlations are calculated between the observations data for other years and the model output for the reference year. Logically, the correlations calculated for the reference year for observations and model outputs would give the better results. By difference with the correlations calculated for other years (with the observations only), we expect to conclude if the model is able to catch the observed variability and for the good reasons. Using this approach, the goal is to give complementary information to those usually obtained when using only scores (correlations, bias, RMSE) calculated for a single year, the studied year. It is thus expected to give additional elements to answer these questions: *Are the performances of the model satisfactory because the model is accurate or just because the model is able to reproduce a situation which is recurrent from year to year? For a given variable, does the model have a good spatial representativeness compared to the corresponding observations?*, and *Are the biases introduced by meteorological or emissions variability or by the formulation of processes in the chemistry-transport model itself?*

The issue to be solved and the tools developed are presented in section 2. The new methodology with the presentation of the indicator developed for this study are presented in section 3. The results and discussions to point out the drivers of model errors are presented in section 4.

## 2 The problem to solve

The problem to solve is presented in a general way by presenting the principle of chemistry-transport modelling. Then, the studied case and the models used are presented.

### 2.1 Regional chemistry-transport modelling

In chemistry-transport modelling, several processes are involved, some of them directly influencing the others. When studying both meteorological and chemical variables, the dependencies between all variables are helpful to know to better interpret the model results. These processes may be broken down into four categories: (i) boundary conditions, (ii) dynamics, (iii) emissions, and (iv) chemistry.

The boundary conditions prescribe the concentrations of chemical species which may enter the simulation domain. Usually for large domains, they are issued from global models as monthly climatologies. They correspond to averaged values suitable to characterize the background concentrations of long-lived species such as ozone, carbon monoxide, mineral dust.

The meteorological variables influence transport and mixing processes, with a direct effect on gas and aerosol plumes locations and their vertical distribution. Cloudiness and temperature impact the photolysis efficiency, the boundary layer height impact the surface mixing of pollutants, rainfall impact the wet deposition. Moreover, meteorology impact emissions: wind variability is the prevalent driver for dust emissions, and it has also a major impact on wildfires emissions. Both temperature and solar irradiance influence the magnitude of biogenic emissions from vegetation. The spatial variability of landuse data has also a strong impact on all these natural emissions.

Anthropogenic emissions are prescribed from databases and the influence of meteorology is limited in the model. Vegetation, fires and mineral dust emissions also depend both on landuse data and meteorology variables. These emissions are difficult to measure, it is almost impossible to quantify their realism.

The chemistry-transport model is a numerical integration tool of all the forcings and processes. The chemical mechanism handles the chemical species life cycle (production and loss) when the deposition processes are the only sink of species. With the model, the spatial (horizontal and vertical) and temporal resolutions are also defined, directly impacting the simulation representativeness and thus the realism of the modelled air pollutant concentrations when they are compared to available observations.

## 2.2 The studied case and the models

The case study focuses on the summer 2013 period (1st May to 31 August) over the Euro-Mediterranean region, this period is called "reference period" in this paper. This case has already been modelled (using WRF and CHIMERE) and the results were discussed in Menut et al. (2015). The same simulation is used in this study, all parameters are identical. The observational data come from different sources depending on the variables, Table 1.

Ozone ($O_3$) and nitrogen dioxide ($NO_2$) are the main pollutants targeted in this study. $PM_{2.5}$, $PM_{10}$ are the surface concentrations of particulate matter with mean mass median diameter lower than 2.5 and $10\mu m$, respectively. Surface concentrations of pollutants are issued from the EBAS database, (Tørseth et al., 2012). AOD and Angström are the Aerosol Optical Depth and the Angström exponent. $T_{2m}$ is the 2m temperature above ground, $U_{10m}$ the wind speed module at 10m above ground and "Precipitation" is the amount of precipitation in millimetres cumulated during a whole day. In this study, all variables are used as daily mean (except for precipitation corresponding to daily cumulated values) in order to (i) have homogeneous scores between the variables, (ii) be able to separate the systematic and the day-to-day variabilities. The use of an hourly time frequency was ruled out to avoid a too strong weight of the diurnal cycle in the temporal variability.

## 3 The proposed methodology

As discussed in the introduction, many scores exist to quantify the model ability to realistically simulate observed pollution events. The correlations scores (temporal and spatial), the Root Mean Squared Error (RMSE) and the bias (the difference between observations and modelled values) are widely used in regional air pollution modelling. The correlations are able to split the relative contributions of systematic meteorology or sources related variability and day-to-day variability. The key point of this study is the study of model variability which is statistically represented by the correlations. The mean bias (or the normalized bias) is not a score able to quantify the variability. And the RMSE is a score containing a part of variability but remains driven by the bias.

The goal of this study is to separate the contributions due to systematic events (*i.e.* when the model seems good, but simulate the same thing every day and every year) and due to sporadic events ((*i.e.* when the model is good because and able to retrieve the day to day variability). This is why the proposed methodology is based on the calculation of the temporal and spatial correlations only.

The methodology follows three steps: (i) compute the correlation scores (spatial and temporal) between the measurements and the model for the whole reference period, (ii) recalculate these scores between the modelled reference period and the observed data for the similar period in 2008, 2009, 2010, 2011 and 2012, (iii) build and use a synthetic score to quantify if the model has high scores for good reasons or not. This is summarized in Figure 1.

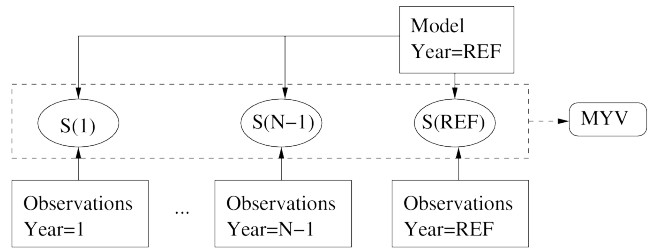

**Figure 1.** *Principle of the multi-year variability score's calculation, using one modelled year and several observations years.*

Of course it seems apparently awkward to evaluate day by day a model with observational data from another year. For

| Variable | Network | Spatial coverage | Vertical coverage | Temporal frequency | Unit |
|---|---|---|---|---|---|
| $O_3$, $NO_2$ | EBAS/EMEP | Europe | Surface | Hourly | ppb |
| $PM_{2.5}$, $PM_{10}$ | EBAS/EMEP | Europe | Surface | Hourly | $\mu g\,m^{-3}$ |
| AOD, Angström | AERONET | Global | Column | Hourly | ad. |
| $T_{2m}$ | BADC | Global | Surface | Tri-hourly | $^oC$ |
| $U_{10m}$ | BADC | Global | Surface | Tri-hourly | $m\,s^{-1}$ |
| Precipitation | BADC | Global | Surface | Tri-hourly | $mm\,day^{-1}$ |

**Table 1.** *List of measurements data used for the statistical comparison with the model results. All data used are issued from surface stations, representative of their own environment. Originally provided hourly or three-hourly, they are used as daily averaged in the present study.*

a given station at a given day of the reference year air concentrations will be affected by a different local meteorology, emissions and also long range transport of chemical species. But we can consider that to take the same date for another year is strictly the same that to choose randomly a date in the same season. This trivial method can emphasize how a model is affected by large scale patterns and long term temporal cycles.

### 3.1 Calculation of the correlation scores

To compute the correlation coefficients, it is important that, for all years of validation, the same list of stations with valid measurements is used. The correlation used in this study is the Pearsons' correlation. Each correlation provides specific information on the quality of the simulation.

The temporal correlation, noted $R_t$, is estimated station by station and using daily averaged data in order to have homogeneous comparisons between all variables. This correlation is directly related to the variability from day to day, for each station.

The $O_{t,i}$ and $M_{t,i}$ represent the observed and modelled values, respectively, at time $t$ and for the station $i$, for a total of $T$ days and $I$ stations. The mean time averaged value $\overline{X_i}$ is:

$$\overline{X_i} = \frac{1}{T}\sum_{t=1}^{T} X_{t,i} \tag{1}$$

The temporal correlation $R_{t,i}$ for each station $i$ is calculated as:

$$R_{t,i} = \frac{\sum_{t=1}^{T}(M_{t,i}-\overline{M_i})\,(O_{t,i}-\overline{O_i})}{\sqrt{\sum_{t=1}^{T}(M_{t,i}-\overline{M_i})^2 \sum_{t=1}^{T}(O_{t,i}-\overline{O_i})^2}} \tag{2}$$

The mean temporal correlation, $R_t$, used in this study is thus:

$$R_t = \frac{1}{I}\sum_{i=1}^{I} R_{t,i} \tag{3}$$

with $I$ the total number of stations. The spatial correlation, noted $R_s$, uses the same formula type except it is calculated from the temporal mean averaged values of observations and model for each location where observations are available. A good correlation shows that the model correctly locates the largest horizontal gradients as known sources and long range transport plumes.

The spatio-temporal mean averaged value is estimated as:

$$\overline{\overline{X}} = \frac{1}{I}\sum_{i=1}^{I}\overline{X_i} \tag{4}$$

and the spatial correlation is thus expressed as:

$$R_s = \frac{\sum_{i=1}^{I}(\overline{M_i}-\overline{\overline{M}})\,(\overline{O_i}-\overline{\overline{O}})}{\sqrt{\sum_{i=1}^{I}(\overline{M_i}-\overline{\overline{M}})^2 \sum_{i=1}^{I}(\overline{O_i}-\overline{\overline{O}})^2}} \tag{5}$$

For the correlations, obviously better scores are expected for the reference year compared to the other. This would confirm that during the transport of pollutants, the model is able to correctly model the day to day variability.

### 3.2 The multi-year variability $I_{mv}$ indicator

The goal of this indicator is to quantify how the correlation between measurements data (for different years) and model output (for the reference year) evolves from a year to another one. We first define the differences, $D$, between all years as:

$$D = \frac{1}{N-1}\left(\sum_{i=1}^{N-1} |s_i - s_N|\right) \tag{6}$$

with $s_N$ the score for the actual year being modelled and $s_i$ the score computed using observations corresponding to other meteorological years (from 1 to $N-1$ if there is $N-1$ other available years for the observations).

We now aim to develop a simple indicator that would follow these rules:

1. The indicator increases with the correlation: More the correlation is high, better the model is.

2. The indicator increases with the differences $D$: more the differences are important more the studied year was different from the others, more the system has a variability.

3. The indicator is moderated if the differences $D$ are low. For example, we want that a correlation of 0.8 has not the same meaning if $D=0$ or $D=1$: the indicator has to give a higher value for ($R=1$, $D=1$) than for ($R=1$, $D=0$).

We can thus estimate a "Multi Year Variability" indicator, noted $I_{mv}$ as:

$$I_{mv} = s_N \left(1 - exp(-D_s)^\delta\right) \tag{7}$$

The value for $\delta$ is arbitrary but it should be larger than unity, in order to have an indicator $I_{mv}$ between 0 and 1. This tuning parameter enables to adapt the relative weight we want to attribute to the absolute value of the scores for the selected year and the differences between all years. In general, we want that a good score for the studied year have a larger weight than the differences between several years. Using $\delta=4$, we consider that the relative weight of the correlation value against the difference reflects well the fact that the model has correct scores and variability.

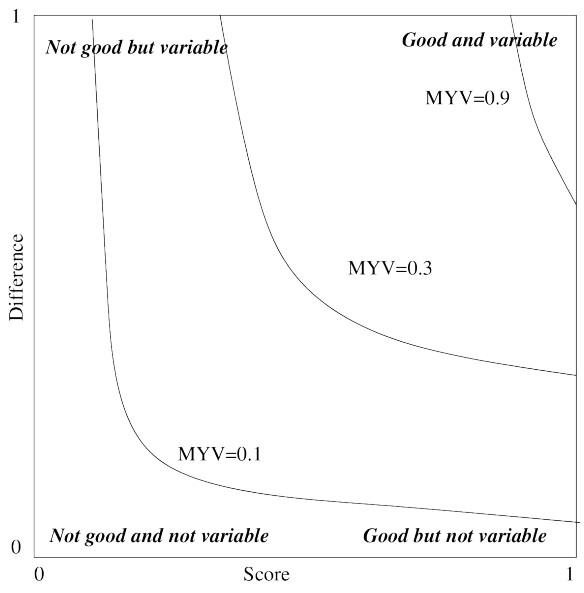

**Figure 2.** *Scheme of the $I_{mv}$ score as a function of the studied year correlation and the multi-years differences.*

The behaviour of $I_{mv}$ is plotted on Figure 2 for values of the scores and the differences ranging from 0 to 1. Ideally we hope that the model performs well for the correlation scores but also be able to reproduce the observed variability. When $I_{mv}$ tends to 1 this means that the correlation value is close to 1 and the differences of the modelled studied year compared to the other years are also close to 1. In reality, this ideal situation is rarely obtained since we are modelling a very complex atmospheric system, based on processes with different variabilities and uncertainties. Moreover, if the correlation is close to zero, the model is definitely poor. Finally, if the difference is close to zero, one can conclude that model performances are independent of the selected year: in that case, $I_{mv}$ is also close to 0.

The role of the indicator $I_{mv}$ is to provide complementary information than the correlation and the differences separately analysed. This indicator has thus to be viewed as complementary to the correlation score and not replacing it. From a subjective point of view, considering the state-of-the art of chemistry-transport modelling and from Figure 2, we consider that the model is accurate and has an acceptable variability for $I_{mv} > 0.3$: this means that the correlation is at least 0.5 and the differences are also at least greater than 0.5. Of course, this value may change if the $\delta$ value is different.

### 3.3 Detailed examples of $I_{mv}$ calculation

To better understand the relevance of $I_{mv}$, two examples are detailed in this section. The scores are calculated for 2m temperature, $T_{2m}$, and for the surface concentration of nitrogen dioxide, $NO_2$. Results are presented in Table 2.

These two variables are presented here because they represent very different variables in a CTM simulation:

- $T_{2m}$ is a meteorological variable, constraining processes both for meteorology and chemistry. Its diurnal cycle is well marked as its latitudinal variability (for large model domains), ensuring a good spatial correlation. In general, it is the less uncertain of modelled meteorological variables.
- $NO_2$ is both a primary and secondary species. Mostly emitted in urbanized areas, the diurnal cycle of this species is well constrained. Depending on meteorological conditions, its lifetime may vary significantly, from hours to days. Modelling this species with CTMs is challenging because several uncertainties are acting at the same time, including the spatial representativeness of the model cell.

| | $T_{2m}$ | | | $NO_2$ | |
|---|---|---|---|---|---|
| Year | $R_s$ | $R_t$ | Year | $R_s$ | $R_t$ |
| 2008 | 0.58 | 0.36 | 2008 | 0.44 | 0.00 |
| 2009 | 0.57 | 0.38 | 2009 | 0.42 | -0.04 |
| 2010 | 0.60 | 0.30 | 2010 | 0.66 | -0.04 |
| 2011 | 0.62 | 0.26 | 2011 | 0.79 | -0.03 |
| 2012 | 0.61 | 0.40 | 2012 | 0.76 | 0.04 |
| 2013 | 0.61 | 0.94 | 2013 | 0.88 | 0.22 |
| D | 0.02 | 0.60 | D | 0.27 | 0.23 |
| $I_{mv}$ | 0.04 | 0.85 | $I_{mv}$ | 0.58 | 0.13 |

**Table 2.** *Scores for $T_{2m}$ and $NO_2$. The correlations are calculated between the observations (2008-2013) and the model results (2013).*

### 3.3.1 Analysis of $T_{2m}$ scores

The spatial correlation is good for all years, ranging from 0.57 (2009) to 0.62 (2011). For the studied year (2013), the score is 0.61, slightly lower than for 2011. Even if the correlation for the selected year is good, it is not significantly better than for the other year, with D=0.02, and this yields to $I_{mv}(R_s)$=0.04. This means that the model reproduces fairly well a spatial pattern that is observed every year. Indeed, the simulation domain is large and the temperature has a latitudinal variability larger than between each measurements stations. This temporal correlation ranges from 0.26 to 0.94. And the best score is for 2013 leading to a good score of $I_{mv}(R_t)$=0.85. The model is thus performing well in capturing the day to day variability for T2m and for the good reasons.

### 3.3.2 Analysis of $NO_2$ scores

Nitrogen dioxide is both a primary and secondary species quickly produced by oxidation of NO and the scores show if the sources are properly placed and if the photochemistry and transport processes have been well simulated. In general, at coarse model resolution, the scores for this species are worse than for ozone. $NO_2$ is very dependent on the quality of emission inventories, however the measurements stations considered in this study are background sites.

The spatial correlation gives a score of $R_s$=0.88 for 2013. Being the best comparison, we obtain $I_{mv}(R_s)$=0.58. This shows the importance of $NO_x$ emission source location that is the main driver of spatial performances. The temporal correlation is low for 2013, $R_t$=0.22, but is close to 0 for other years. In the end, we have a low score with $I_{mv}(R_t)$=0.13 even if the simulated year is better. These two scores show that the model certainly captures the right location of emission sources (low variability of $R_s$). For the temporal variability, the model is not able to reproduce the day to day variability, but it remains significantly better for the reference year compare to the others.

## 4 Results and discussion

The correlations are calculated for all variables described in Table 1 and for the years 2008 to 2013, it is reminded that only the May to August 2013 period was modelled. Results are presented as time series in Figure 3. Using all correlations and differences values, a $I_{mv}$ is estimated for each variable. Results (Table 3) are discussed in the following sections.

### 4.1 Meteorological variables

Scores for $T_{2m}$ were discussed in the previous section. The calculation of $u_{10m}$ also gives satisfactory results with $R_t$=0.60 and $I_{mv}$=0.54. The spatial correlation, $R_s$=0.09, is not correct and very variable from one year to another, lead-

| Variable | $R_s$ | | | $R_t$ | | |
|---|---|---|---|---|---|---|
| | Value | D | $I_{mv}$ | Value | D | $I_{mv}$ |
| $T_{2m}$ | 0.61 | 0.02 | 0.04 | 0.94 | 0.60 | **0.85** |
| $u_{10m}$ | 0.09 | 0.09 | 0.03 | 0.60 | 0.56 | **0.54** |
| precip | 0.78 | 0.29 | **0.54** | 0.30 | 0.31 | 0.21 |
| AOD | 0.97 | 0.02 | 0.09 | 0.45 | 0.34 | **0.33** |
| ANG | 0.91 | 0.04 | 0.14 | 0.59 | 0.44 | **0.49** |
| $O_3$ | 0.69 | 0.13 | 0.29 | 0.32 | 0.27 | 0.21 |
| $NO_2$ | 0.88 | 0.27 | **0.58** | 0.22 | 0.23 | 0.13 |
| $PM_{2.5}$ | 0.16 | 0.15 | 0.07 | 0.27 | 0.32 | 0.20 |
| $PM_{10}$ | 0.57 | 0.43 | **0.47** | 0.11 | 0.10 | 0.04 |
| Ammonium | 0.20 | 0.13 | 0.08 | 0.21 | 0.20 | 0.12 |
| Sulphate | 0.51 | 0.21 | 0.29 | 0.31 | 0.34 | 0.23 |
| Nitrate | 0.15 | 0.51 | 0.13 | 0.09 | 0.08 | 0.03 |

**Table 3.** *The $I_{mv}$ values for all variables: the meteorology with $T_{2m}$, $u_{10m}$ and precipitation rate, the vertically integrated column of aerosols with the Aerosol Optical Depth (AOD) and the Angström exponent (ANG), the surface concentrations of all aerosols in term of size distribution with $PM_{2.5}$ and $PM_{10}$ and for the inorganic species with $D_p < 10\,\mu m$. Values of $I_{mv}$ above 0.3 are bolded. Units of the variables are detailed in Table 1.*

ing to $I_{mv}$=0.03. As for $T_{2m}$, we also have an effect of the model resolution and the representativeness of the variable. Scores for the precipitation are correct, with a very good spatial correlation leading to $I_{mv}(R_s)$=0.54. For the day to day variability, the score is less good with $I_{mv}(R_t)$=0.21 but significantly higher for 2013. These scores showed that the meteorological forcing is well retrieved, and better for the year being considered compared to other years.

### 4.2 Optical properties

The optical properties are directly linked to the atmospheric composition of aerosol and may be quantified using the Aerosol Optical Depth (AOD) and the Angström exponent (ANG).

For the AOD, the spatial correlation is very good for 2013, $R_s$=0.97 but it is as good or better for other years. This means that we model a rather recurring phenomenon: every year the same stations are on average exposed to aerosol plumes: $I_{mv}(R_s)$=0.09. The temporal correlation is lower with $R_t$=0.45 but much better than for other years: $I_{mv}(R_t)$=0.33. This means that the model partly reproduced the observed temporal variability but the events are changing from one year to another and the model captures well these changes. The AOD are sensitive to desert dust outbreaks in summer in that region. This means that large scale systems are driving the aerosol plumes; they are spatially recurrent and temporally better estimated for the year being considered than for other years.

For the ANG, the spatial correlation is very good, $R_s$=0.91 but also persistent leading to a low score of $I_{mv}(R_s) = 0.14$.

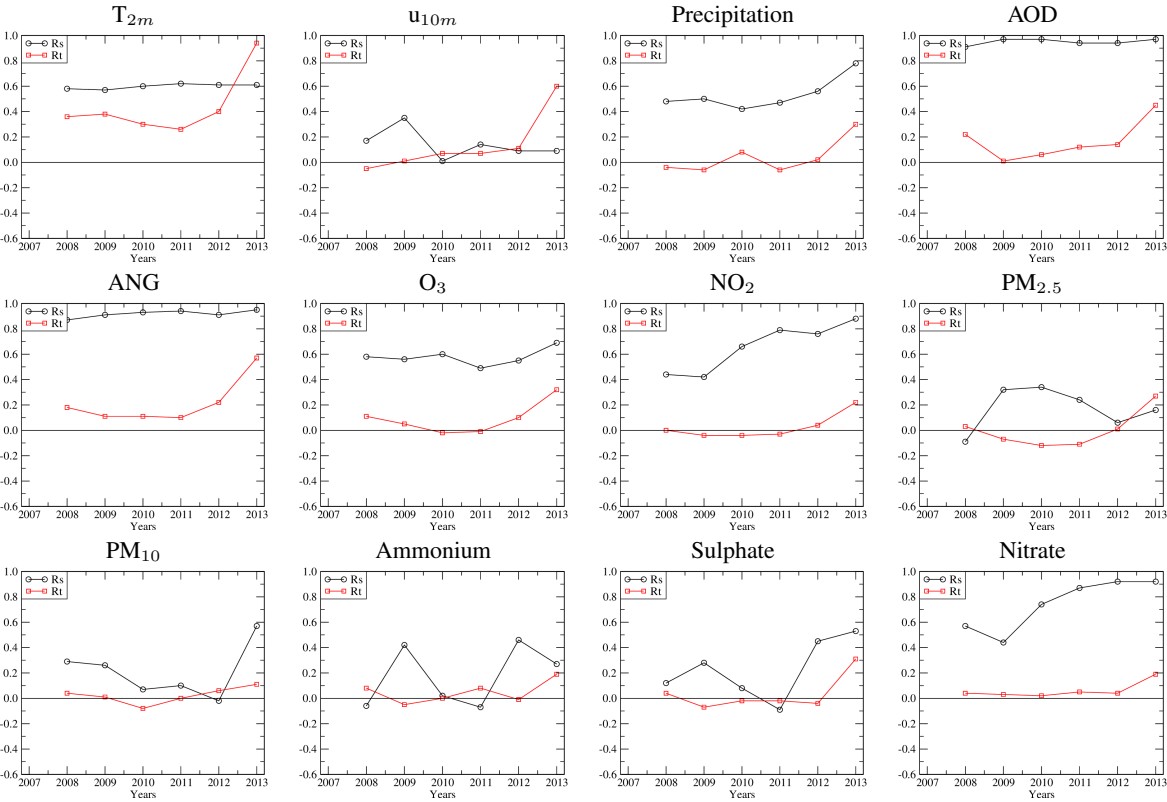

**Figure 3.** *Multi years scores for $T_{2m}$, $u_{10m}$, the precipitation rate, Aerosol Optical Depth (AOD), Angström exponent (ANG), surface concentrations of $O_3$, $NO_2$, $PM_{2.5}$, $PM_{10}$, Ammonium, Sulphate and Nitrate. The correlations are calculated between the observations (2008-2013) and the model results (2013). The spatial correlation, $R_s$, is in black and the temporal correlation, $R_t$ is in red.*

The temporal correlation is much better for 2013 than other years with $I_{mv}(R_t)$ = 0.49. This is probably due to a size distribution that is not necessarily well simulated from one day to another (showed by AOD) but the relative contributions of fine and coarse aerosol atmospheric load are fairly reproduced. This feature highlights the high sensitivity of the AOD calculation to the modelled aerosol size distribution, although the overall mass emitted and transported could be realistic.

Globally, the AOD and ANG reflect the model's ability to retrieve the long range transport of long-lived aerosols which depends on several processes (emissions, transport, and deposition). These scores show the model is able to retrieve these yearly recurrent plumes but the model size distribution of particles clearly requires improvements.

### 4.3 Surface concentrations

The spatial correlation is good for $O_3$, $NO_2$ and $PM_{10}$, with $R_s$=0.69, 0.88 and 0.57 respectively. For $PM_{2.5}$ this correlation is low with $R_s$=0.16. The $PM_{10}$ shows that the largest particles are well modelled over the whole domain, and this was also the conclusion for the AOD and ANG. The low score for $PM_{2.5}$ indicates that for the aerosol distribution, the

fine mode is less well modelled than the coarse mode. This is confirmed by the scores of the aerosol inorganic species, Ammonium, Sulphate and Nitrate. Except for Sulphate (with $R_s$=0.51), the spatial correlations are 0.15 for Nitrate and 0.20 for Ammonium. Thus, the fine part of the aerosol is not well modelled mainly due to a deficiency in the modelling of nitrates.

The temporal correlations have a completely different behaviour that the spatial correlations. The values are generally low, from $R_t$=0.09 for Nitrate to $R_t$=0.32 for $O_3$. Surprisingly, the $PM_{10}$ concentrations display a good spatial correlation but a poor temporal correlation. This is due to the long lifetime in the atmosphere of non-reactive species such as mineral dust: large plumes are correctly modelled over regions but the day to day variability needs improvements. Another point is the good spatial correlation for $NO_2$ (and for the good reasons with $I_{mv}$=0.58) but its low temporal correlation with $R_t$=0.22 and a low $I_{mv}$=0.13. In this case, this means we have a correctly localized anthropogenic emissions inventory (main source of $NO_2$) but difficulties to model the day to day chemistry.

In conclusion for the surface concentrations, we can conclude that $O_3$, $NO_2$ and $PM_{10}$ concentrations are spatially

well modelled and this is not due to a recurrent behaviour, $I_{mv}$ having high values. For particles, the problem is more related to the fine mode, where PM$_{2.5}$ concentrations are not well located. This modelling problem is highlighted by the low correlations and $I_{mv}$ values for the inorganic species. For the temporal correlations, the scores are always lower than for the spatial correlation but also always higher for the reference year than for the other years.

### 4.4    Representation of results on a single plot

Complementary to the Table 3, Figure 4 reports the results on a single plot. The x-axis represents the correlation (spatial or temporal), the y-axis represents the differences between all years, D. For each studied variables, their values are reported on the Figure where the colours represent the value of $I_{mv}$. The interpretation of these results follows the quality criteria presented in the academic scheme in Figure 2.

This presentation shows an important spread for the spatial correlation results. If the relative differences $D$ range from 0 to 0.6, the correlations range from 0.09 (for the 10m wind speed) to 0.97 (for AOD). The common point is that there is no variable with differences above 0.5. This means that, spatially, the studied problem shows systematic patterns from year to year. The low values of correlations show that some variables are systematically badly estimated. This means that some meteorological structures (for $u_{10m}$) or emission sources (contributing to the PM$_{2.5}$ surface concentrations) are systematically mis-located.

The representation of temporal correlations shows a specific linear pattern. The largest correlation values are positively correlated with differences. This temporal correlation represents the day to day variability at each location. This means that the studied problem is based on high day to day variability without similar consecutive days (in this case, one would have high correlations but low differences). This illustrates the fact that the studied problem is primarily an issue of sporadic events and the model is able to correctly find this variability from one day to another.

## 5    Conclusions

At first glance, using a different year than the simulated one for the day to day evaluation seems awkward. However, we can learn more about the performances of chemistry transport models than using a single statistical indicator. Of course, this approach will never replace a strict evaluation of a pollution case analysis using time series, vertical profiles and usual error statistics. However, it offers a very fast and integrated vision of the strengths and weaknesses of a model with very little calculation. This methodology can also be deployed in inter-comparison exercises.

To answer the questions presented in the introduction, and for this particular model and simulated period, the following

conclusions can be drawn. The model always simulates better the studied year than any other meteorological year and it is able to reproduce the day to day variability for high concentrations of pollutants.

The spatial correlation is good for 2m temperature and precipitation rate, but not for wind speed: this highlights the fact that the modelled domain is large and the resolution not optimized for small scale processes. The spatial correlation is also very good for the long-range transport of particles as demonstrated with R$_s$=0.96 and 0.90 for AOD and ANG. But, since this feature occurs every year, this leads to low $I_{mv}$ values. This means that for a large domain, the main spatial patterns of particle concentrations are recurrent and well modelled. The chemical species that are best modelled are either species with a long atmospheric lifetime (PM$_{10}$) or species well spatially constrained on the domain (such as NO$_2$ mainly due to anthropogenic emissions). For particles, the results depend on the size distribution: the largest particles are better simulated than the finest ones.

The conclusions are different for the temporal correlation. The scores are calculated using daily observations and modelled outputs. Thus, these scores reflect the ability of the model to retrieve the day to day variability. As for the spatial correlation, scores are good for the meteorological variables. For the aerosol, and mainly for the long-lived species (such as mineral dust), the temporal correlation is also correct as the $I_{mv}$ values: $I_{mv}$=0.33 and 0.49 for AOD and ANG respectively. But for the short-live species the temporal correlation and the $I_{mv}$ values are low. This means that improvements are required in priority for the day to day variability compared to the locations of emissions. This may probably be due to the atmospheric transport, the spatial variability of 10m wind speed being poorly simulated. But, on overall, the temporal correlation is better for the studied year than for the others, showing that the problem is highly variable from year to year, but the model is significantly able to catch the evolution of the atmospheric composition.

*Acknowledgements.* This study is partly funded by the French Ministry in charge of Ecology.

## 6    Code and/or data availability

This study presenting a methodology using existing data and models, all required information are already included in this article.

## References

Appel, K. W., Gilliam, R. C., Davis, N., Zubrow, A., and Howard, S. C.: Overview of the atmospheric model evaluation tool (AMET) v1.1 for evaluating meteorological and air quality models, Environmental Modelling and Software, 26, 434 – 443, doi:doi.org/10.1016/j.envsoft.2010.09.007, 2011.

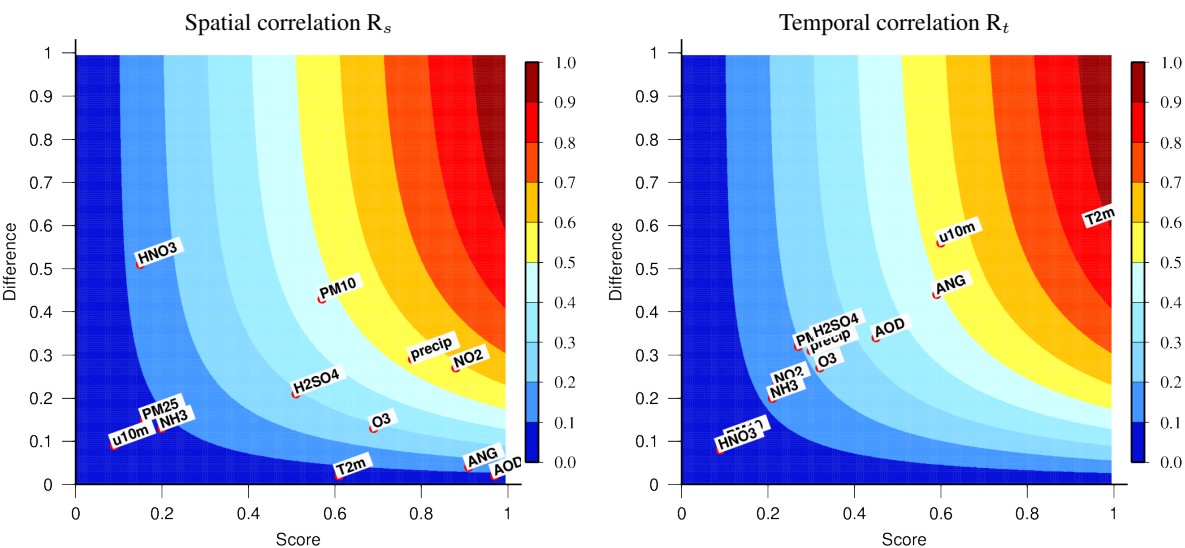

**Figure 4.** *Results of the $I_{mv}$ scores for the spatial and temporal correlations. For each model variable its value is represented using the correlation on the x-axis and the difference between the studied year and the others on the y-axis. The colours represent the $I_{mv}$ values.*

Baldridge, K. and Cox, W.: Evaluating air quality model performance, Environmental Software, 1, 182 – 187, doi:doi.org/10.1016/0266-9838(86)90023-7, 1986.

Bessagnet, B., Pirovano, G., Mircea, M., Cuvelier, C., Aulinger, A., Calori, G., Ciarelli, G., Manders, A., Stern, R., Tsyro, S., García Vivanco, M., Thunis, P., Pay, M.-T., Colette, A., Couvidat, F., Meleux, F., Rouïl, L., Ung, A., Aksoyoglu, S., Baldasano, J. M., Bieser, J., Briganti, G., Cappelletti, A., D'Isidoro, M., Finardi, S., Kranenburg, R., Silibello, C., Carnevale, C., Aas, W., Dupont, J.-C., Fagerli, H., Gonzalez, L., Menut, L., Prévôt, A., Roberts, P., and White, L.: Presentation of the EURODELTA III intercomparison exercise - evaluation of the chemistry transport models' performance on criteria pollutants and joint analysis with meteorology, Atmospheric Chemistry and Physics, 16, 12 667–12 701, doi:10.5194/acp-16-12667-2016, http://www.atmos-chem-phys.net/16/12667/2016/, 2016.

Campbell, P., Zhang, Y., Yahya, K., Wang, K., Hogrefe, C., Pouliot, G., Knote, C., Hodzic, A., Jose, R. S., Perez, J. L., Guerrero, P. J., Baro, R., and Makar, P.: A multi-model assessment for the 2006 and 2010 simulations under the Air Quality Model Evaluation International Initiative (AQMEII) phase 2 over North America: Part I. Indicators of the sensitivity of O3 and PM2.5 formation regimes, Atmospheric Environment, 115, 569 – 586, doi:doi.org/10.1016/j.atmosenv.2014.12.026, 2015.

Chang, J. and Hanna, S.: Air quality model performance evaluation, Meteorology and Atmospheric Physics, 87, 167–196, doi:10.1007/s00703-003-0070-7, 2004.

Cox, W. M. and Tikvart, J. A.: A statistical procedure for determining the best performing air quality simulation model, Atmospheric Environment. Part A. General Topics, 24, 2387 – 2395, doi:doi.org/10.1016/0960-1686(90)90331-G, 1990.

Galmarini, S., Bianconi, R., Appel, W., Solazzo, E., Mosca, S., Grossi, P., Moran, M., Schere, K., and Rao, S.: {ENSEMBLE} and AMET: Two systems and approaches to a harmonized, simplified and efficient facility for air quality models develop-

ment and evaluation, Atmospheric Environment, 53, 51 – 59, doi:doi.org/10.1016/j.atmosenv.2011.08.076, aQMEII: An International Initiative for the Evaluation of Regional-Scale Air Quality Models - Phase 1, 2012.

Kioutsioukis, I. and Galmarini, S.: *De praeceptis ferendis*: good practice in multi-model ensembles, Atmospheric Chemistry and Physics, 14, 11 791–11 815, doi:10.5194/acp-14-11791-2014, 2014.

Kopacz, M., Jacob, D. J., Fisher, J. A., Logan, J. A., Zhang, L., Megretskaia, I. A., Yantosca, R. M., Singh, K., Henze, D. K., Burrows, J. P., Buchwitz, M., Khlystova, I., McMillan, W. W., Gille, J. C., Edwards, D. P., Eldering, A., Thouret, V., and Nedelec, P.: Global estimates of CO sources with high resolution by adjoint inversion of multiple satellite datasets (MOPITT, AIRS, SCIAMACHY, TES), Atmospheric Chemistry and Physics, 10, 855–876, doi:10.5194/acp-10-855-2010, 2010.

Lemaire, V. E. P., Colette, A., and Menut, L.: Using statistical models to explore ensemble uncertainty in climate impact studies: the example of air pollution in Europe, Atmospheric Chemistry and Physics, 16, 2559–2574, doi:10.5194/acp-16-2559-2016, http://www.atmos-chem-phys.net/16/2559/2016/, 2016.

Mao, Y. H., Li, Q. B., Henze, D. K., Jiang, Z., Jones, D. B. A., Kopacz, M., He, C., Qi, L., Gao, M., Hao, W.-M., and Liou, K.-N.: Estimates of black carbon emissions in the western United States using the GEOS-Chem adjoint model, Atmospheric Chemistry and Physics, 15, 7685–7702, doi:10.5194/acp-15-7685-2015, 2015.

Marécal, V., Peuch, V.-H., Andersson, C., Andersson, S., Arteta, J., Beekmann, M., Benedictow, A., Bergström, R., Bessagnet, B., Cansado, A., Chéroux, F., Colette, A., Coman, A., Curier, R. L., Denier van der Gon, H. A. C., Drouin, A., Elbern, H., Emili, E., Engelen, R. J., Eskes, H. J., Foret, G., Friese, E., Gauss, M., Giannaros, C., Guth, J., Joly, M., Jaumouillé, E., Josse, B., Kadygrov, N., Kaiser, J. W., Krajsek, K., Kuenen, J., Kumar, U., Liora, N., Lopez, E., Malherbe, L., Martinez, I.,

Melas, D., Meleux, F., Menut, L., Moinat, P., Morales, T., Parmentier, J., Piacentini, A., Plu, M., Poupkou, A., Queguiner, S., Robertson, L., Rouïl, L., Schaap, M., Segers, A., Sofiev, M., Tarasson, L., Thomas, M., Timmermans, R., Valdebenito, A., van Velthoven, P., van Versendaal, R., Vira, J., and Ung, A.: A regional air quality forecasting system over Europe: the MACC-II daily ensemble production, Geoscientific Model Development, 8, 2777–2813, doi:10.5194/gmd-8-2777-2015, http://www.geosci-model-dev.net/8/2777/2015/, 2015.

Menut, L.: Adjoint modelling for atmospheric pollution processes sensitivity at regional scale during the ESQUIF IOP2, J. Geophys. Res., 108, 8562, doi:10.1029/2002JD002549, 2003.

Menut, L., Mailler, S., Siour, G., Bessagnet, B., Turquety, S., Rea, G., Briant, R., Mallet, M., Sciare, J., Formenti, P., and Meleux, F.: Ozone and aerosol tropospheric concentrations variability analyzed using the ADRIMED measurements and the WRF and CHIMERE models, Atmospheric Chemistry and Physics, 15, 6159–6182, doi:10.5194/acp-15-6159-2015, http://www.atmos-chem-phys.net/15/6159/2015/, 2015.

Pison, I., L.Menut, and G.Bergametti: Inverse modeling of surface NOx anthropogenic emissions fluxes in the Paris area during the ESQUIF campaign, Journal of Geophysical Research, Atmospheres, 112, D24 302, doi:10.1029/2007JD008871, 2007.

Porter, P. S., Rao, S. T., Hogrefe, C., Gego, E., and Mathur, R.: Methods for reducing biases and errors in regional photochemical model outputs for use in emission reduction and exposure assessments, Atmospheric Environment, 112, 178 – 188, doi:doi.org/10.1016/j.atmosenv.2015.04.039, 2015.

Prank, M., Sofiev, M., Tsyro, S., Hendriks, C., Semeena, V., Vazhappilly Francis, X., Butler, T., Denier van der Gon, H., Friedrich, R., Hendricks, J., Kong, X., Lawrence, M., Righi, M., Samaras, Z., Sausen, R., Kukkonen, J., and Sokhi, R.: Evaluation of the performance of four chemical transport models in predicting the aerosol chemical composition in Europe in 2005, Atmospheric Chemistry and Physics, 16, 6041–6070, doi:10.5194/acp-16-6041-2016, 2016.

Simon, H., Baker, K., and Phillips, S.: Compilation and interpretation of photochemical model performance statistics published between 2006 and 2012, Atmospheric Environment, 61, 124–139, doi:10.1016/j.atmosenv.2012.07.012, 2012.

Solazzo, E. and Galmarini, S.: Comparing apples with apples: Using spatially distributed time series of monitoring data for model evaluation, Atmospheric Environment, 112, 234 – 245, doi:doi.org/10.1016/j.atmosenv.2015.04.037, 2015.

Solazzo, E. and Galmarini, S.: Error apportionment for atmospheric chemistry-transport models - a new approach to model evaluation, Atmospheric Chemistry and Physics, 16, 6263–6283, doi:10.5194/acp-16-6263-2016, 2016.

Thunis, P., Pederzoli, A., and Pernigotti, D.: Performance criteria to evaluate air quality modeling applications, Atmospheric Environment, 59, 476 – 482, doi:doi.org/10.1016/j.atmosenv.2012.05.043, 2012.

Tørseth, K., Aas, W., Breivik, K., Fjæraa, A. M., Fiebig, M., Hjellbrekke, A. G., Lund Myhre, C., Solberg, S., and Yttri, K. E.: Introduction to the European Monitoring and Evaluation Programme (EMEP) and observed atmospheric composition change during 1972-2009, Atmospheric Chemistry and Physics, 12, 5447–5481, doi:10.5194/acp-12-5447-2012, http://www.atmos-chem-phys.net/12/5447/2012/, 2012.

Valari, M. and Menut, L.: Does increase in air quality models resolution bring surface ozone concentrations closer to reality?, Journal of Atmospheric and Oceanic Technology, doi:10.1175/2008JTECHA1123.1, 2008.

Vautard, R., Moran, M. D., Solazzo, E., Gilliam, R. C., Matthias, V., Bianconi, R., Chemel, C., Ferreira, J., Geyer, B., Hansen, A. B., Jericevic, A., Prank, M., Segers, A., Silver, J. D., Werhahn, J., Wolke, R., Rao, S., and Galmarini, S.: Evaluation of the meteorological forcing used for the Air Quality Model Evaluation International Initiative (AQMEII) air quality simulations, Atmospheric Environment, 53, 15 – 37, doi:doi.org/10.1016/j.atmosenv.2011.10.065, aQMEII: An International Initiative for the Evaluation of Regional-Scale Air Quality Models - Phase 1, 2012.