# Peer review of "An alternative way to evaluate chemistry-transport models variability"

_Geoscientific Model Development, 2016_

## Referee Comment (RC1) · Anonymous Referee #1 · 3 Aug 2016

In this paper, the authors present an extension of the evaluation of (atmospheric chemistry) models by using measurements from other years than the year which was simulated by the model. New scores are introduced to quantify the ability of the model to capture the day to day variability as opposed to persistent patterns.

General comments:

While reading the paper I asked myself the question if the approach presented by the authors has a real added value as compared to a more traditional model evaluation based on bias, RMS and (one type of) correlation, and may be adopted by other groups. In the end I decided that it probably does, for the following reasons:

- The approach proposed quantifies the importance of day-to-day, weather dominated variability versus systematic patterns which are repeated from year to year.

[Figure]

- The approach naturally leads to an overview of the performance for multiple species in one graph (e.g. Fig.5), which is especially also useful (maybe even more useful) for comparisons between different models. This include both trace species as well as meteorological variables. This is a bit similar to the use of Taylor diagrams.

- The approach explicitly exploits both spatial and temporal correlations, which bring complementary information.

- The approach provides new insight into the performance of the WRF-CHIMERE model.

Because of this I am in favour of publication. However, to my opinion there are several major and minor points to be addressed before the paper can be considered by GMD. These are listed below:

- Is this approach really new? The authors provide a few interesting references in the paper, but I would like to see a more systematic overview of the model evaluation approaches and techniques/scores adopted in the past (e.g. including several European/American CTM intercomparison exercises) to better understand the added value of the approach proposed.

- The formulation is incomplete, and mathematical formulas are not well defined. In particular, the authors should provide the equations for $R_s$ and $R_t$, and the mathematical formula for the MYV needs more discussion, see my comments below. Also, the authors should motivate why the $R_{s,t}$ scores are chosen.

- The MYV is not really a model score to my opinion, but rather an indicator of how much the score is influenced by day-to-day variability. In particular one can argue that R=1 and D=0 is a good result. Also, I wonder if a formula for MYV is really needed. Showing D and R is maybe enough (see e.g. Fig. 5)? This should be more carefully presented/discussed.

Detailed comments:

p4, l13: "they are used as daily averaged in the present study": why this choice to focus on daily averages instead of hourly values? Please motivate.

p4, table: Provide also the full names of the variables, e.g. "Temperature at 2m above ground" etc.

p4, last line: replace "same day for another is" by "same day for another year is"

p5, l4: "The correlation is the more appropriate statistical metric for such analysis." Please explain and motivate this statement in detail. This is important for the rest of the paper!

p5, l8: "The spatial correlation, noted Rs, is calculated from the temporal mean averaged values of observations and model for each location where observations are available." Please provide a detailed mathematical formula/recipe to be clear. Are observations and model first collocated for individual observations, or are means computed and then compared. Are these means of daily means or means of hourly values? It is important to define precisely how the correlations are computed: the devil is in the details.

p5, 13: Also for the temporal correlation: be more precise. Is it based on daily means, hourly values or something else.

p5, l14: "The longer the atmospheric lifetime of the species, the lower the relevance of temporal correlation" I would dispute this. For long-lived tracers the transport (wind direction) and location/strength of the sources becomes crucial, directly influencing temporal correlations. I suggest to remove this remark.

p5, eq.1: Why is there an absolute value introduced. Instead of absolute(s_i-s_N) I would suggest (S_N minus s_i) assuming higher values of "s" (or "s" close to 1) indicate better performance, which is the case for correlations.

p6, eq.2: Remove the "X" (multiplication) from the formula. This is not needed (in eq.1 there is also no X). Please introduce a one character symbol for the "Multi Year

Variability" instead of writing "MYV" in eq 2, which, in mathematical formula's means M times Y times V. "D_s" has not been introduced: is it the same as "D" ?

p6, eq.2: Why this complicated exponential form?? It seems that you ideally would have the MYV to be =1 for (s=1 and D=1), and =0 for (s=0 or D=0). A much simpler form s_{MYV} = s_N D would do the trick. In fact, eq.2 is not =1 for s_N=D=1. Where does this formula come from? Is there a reference to a paper introducing this form? Also, it would be good if the formula has clear limits, e.g. 0 (very bad) and 1 (very good). This is not the case when D=1.

p7, l2: Where does the number 0.3 come from? It will depend a lot on how the score "s" is defined. The number seems arbitrarily chosen.

p7, l14: " . . . is challenging because several uncertainties . . . "

Table 2: It would be helpful to remind the reader that these are Summer periods (1-5 to 1-9) and that the scores are based on daily mean values. Please also highlight the special situation for 2013 (I would suggest to start with 2013, add a thick line, and continue with 2008 2009 . . . Perhaps it can be stressed once more in the caption that observations for 2008-2012 (and 2013) are compared with 2013 model results.

Figure 4: Caption is incomplete.

Table 3: ". . .Values of MYV above 0.3 are shown in bold. . ."

p11, l18: . . .with differences above 0.5. . . .

---

## Referee Comment (RC2) · Anonymous Referee #2 · 19 Oct 2016

This work addresses the important issue of the validation of chemistry transport models. The authors present a new methodology in which the traditional approach consisting of comparing measurements with model results for a given time period is extended to comparisons of the same model results with measurements from other years. The authors develop then a specific indicator on this basis that allows discriminating results that are good for the good reason from those that are good only because of highly persistent pattern present in the observations from year to year. While the proposed methodology is original and has a potential to complement the traditional approach, the authors remain unfortunately superficial and qualitative in their way of presenting and applying this methodology. As a consequence, the proposed examples are qualitative as well and are not helpful. Finally, the document is poorly written: (1) English would need revisions throughout the whole document and (2) many sections would need to

be re-written (some suggestions are proposed below).

Major points:

1) The authors mention Solazzo and Galmarini (2016) for their decomposition of the error but they finally focus on the correlation only. As noted by these two Authors but also by many others (the referencing to other works relating to model evaluation should be improved), it is important to look at all three possible source of errors because focusing on the only correlation may lead to the wrong conclusions (see comments below). I'm wondering why the Authors make this choice as the proposed methodology could easily be developed for other indicators that are more representative of the overall model performance (e.g. MSE).

2) The approach proposed by the Authors remains qualitative and the interpretation depend on the setting of an arbitrary threshold (e.g. MYV=0.3 in Figure 3). Throughout the text, the Authors make qualitative judgements (0.6 is good, 0.5 is poor...). This limits the usefulness of the proposed methodology as we never know what a good value of the indicator is. I do not understand this limitation as it would seem relatively straightforward to calculate a value of the MYV indicator in a similar way but on the only basis of measurements. This observation-based MYV value could then serve as the threshold beyond which model results would be considered good enough.

3) The document is poorly written. Many sections are unclear and lack sufficient details to be understood. Some suggestions are provided below but the whole document should be thoroughly revised.

Minor points:

1) P1, l1: The title is not very representative of the work

2) P1, l3: "and by natural" –> "and natural"

3) P1, l19: the transport

4) P1, l20: or from the QAERONET

5) P2, l1: can be

6) P2, l2-3: sentence to be revised

7) P2, l4: "spatial representativeness" –> "spatial representativeness of the monitoring stations". In addition, this concept is mentioned for the first time and should be defined. Finally, I do not get the added value of mentioning this here.

8) P2, l5: "to isolate problems intrinsic to the models,". This is unclear and should be re-phrased

9) P2, l6: "relevant": which ones?

10) P2, l7: "but often with huge" –> "but often require important"

11) P2, l8: references should be within brackets

12) P2, l15-17 & l18-20: if the authors cite these works, they should explain in a little bit more detail their main aspects and why these are important in the context of their work. All these references are introduced independently from the scope of the work. For example on l18, what is the decomposition about? L17, what did Rea et al. find that is relevant for this work. . .

13) P2, l18: scores is often misused in the text. Sometimes as real score, some times meant as correlation. I guess the authors here refer to indicators.

14) P2, l23: "we apply these scores to a model simulation" is unclear. I do not understand how to apply a score to a model simulation. Please check all occurrences of "scores" and check relevance.

15) P2, l27: provide

16) P2, l29: spatial representativeness is not yet defined. Is special representativeness really assessed by this method? I do not believe so (see following comments)

17) P2, l33: Score meant as indicator?

18) P3, figure 1: I do not believe this figure helps understanding. The proposed methodology is quite universal and does not require to enter these details

19) P3, l7: forcings

20) P3, l9-23: these lines are not necessary to the methodology and application

21) P4, l4: unclear

22) P4, l9: for –> in

23) P4, l12: variable (Table 1)

24) P4, l16: and during –> for

25) P4, l21: take the same day for another –> to re-phrase

26) P5, l4: why is correlation the more appropriate metric. Why couldn't we say the same for the bias, for example?

27) P5, l5: What is a usual correlation score? A correlation is a correlation and a score a score!

28) P5, l11-12: I disagree with the authors. A good correlation score does not indicate that the resolution is adequate, transport is adequate . . . Correlation could be 1 while keeping a huge bias due to a too coarse resolution.

29) P5, l16: "particularly": why?

30) P5, l20: which differences? Between what?

31) P6, l5: why should it be larger than unity?

32) P6, l5-6: These lines are totally unclear and should be re-phrased

33) P6, l7: have –> has

34) P6, l7: why do we want that a good score...": although it may appear straightforward, please give a few words of explanation.

35) P6, l9: What is an academic value of the score, what is the score meaning here?

36) P6, l10: absolute score but also variable: unclear

37) P6, l9-15: this all paragraph is unclear and should be rewritten

38) P6, l18-19: 5 times scores in these sentences!

39) Figure 3 and Figure 6 seems to be inconsistent in terms of X axis labeling.

40) P7, l1: from Figure 3

41) P7, l1: we can consider that

42) P7, l1-2: This means that all conclusions will remain subjective because of this arbitrarily fixed delta parameter. I believe that a measurement based threshold value for delta can be fixed, withdrawing this arbitrary aspect (see major comment above).

43) P7, l6: done –> calculated

44) P7, l6: MYV scores

45) P7, l12: vary a lot –> vary significantly

46) P7, l13: is challenging because

47) P7, l13: again spatial representativeness needs to be defined

48) P7, l17: "The spatial correlation is good for all years". I do not understand which arguments the Authors use to state that the score is good. If the spatial pattern is easy to reproduce, it could well be that a correlation of 0.7 should be considered as bad. This seems to be confirmed by the next sentence: "the model reproduces fairly well a spatial patter observed every year". One way forward is to calculate the correlations on the only basis of measurements to get some indicative threshold of what is good or

not.

49) P8, l2: Are we sure this is for the good reasons?

50) P8, l6: "This species is secondary" seems to contradict p7, l12.

51) P8, l6,7: I do not agree that a good score for correlation is indicating a good transport, photochemistry...Correlation is indeed only one of the indicators to assess model performances and it only provides a partial vision of model performances. Correlation could be perfect even with a very large bias.

52) P8, l8: low –> coarse

53) P8, l8: less good –> worse

54) "Its spatial extent of its representativeness": totally unclear, this should be rephrased

55) P8, l18: "The scores": The correlations are calculated, not the scores which are the correlation values

56) P8, l20: "each score type". I do not understand what the Authors mean.

57) P8, l20: "Results are presented in Table 3. These results..." –> Results (Table 3) are discussed...

58) P8, l24: why only?

59) P8, l24: Which arguments are used to state that the spatial correlation is not correct?

60) P8, l24: for one year –> from one year

61) P8, l26, 27 and 28: "very good spatial", "less good", "well retrieved". The Authors should explain how they come to these statements.

62) P8, l31: A few words to explain what the AOD and ANG are would be helpful

63) Figure 4 caption: Should include explanations of the two curves represented

64) P10, l9,10,11: Again I do not agree with these conclusions which cannot be drawn from the only correlation values.

65) P11, l19-20: this sentence is unclear

66) P12, l29: dued –> due

---

## Author Comment (AC1) · 19 Dec 2016

**Review of paper gmd-2016-153**
**An unusual way to validate regional chemistry-transport models, L. Menut et al.**

Dear Editor and reviewers,

We acknowledge the reviewers for the time spent to evaluate our work. We also acknowledge the Editor and we made all proposed changes in the revised manuscript.

There is some common remarks which can be synthesized:

1. *The bibliography could be improved*: this was done and the state of the art regarding the current ways to validate CTMs was rewritten. In brief, the following references were added: [Baldridge and Cox, 1986], [Cox and Tikvart, 1990], [Chang and Hanna, 2004] [Appel et al., 2011], [?], [Galmarini et al., 2012], [Vautard et al., 2012], [Bennett et al., 2013], [Schaap et al., 2015], [Campbell et al., 2015], [Bessagnet et al., 2016].

2. *The scores could use RMSE and bias*: This is right, and in fact, we did it during the preparation of the manuscript. This was removed for the submission because we considered that the added-value was low. A long explanation for this choice is proposed in the answer to the reviewer #2.

3. *The interest to have a MYV score*: There is two kinds of novelties in this paper. First, the fact to use data from other years than the studied year is the most important novelty. This is why the title is "unusual way", because this is the first time that such way to estimate the model realism is used. Second, the MYV score. This is also new and the goal is to have a quantified link between the "differences" and the scores (correlation, RMSE, etc.). The constant value is arbitrary, this is true. But the user can select another value. In the case of CTM, this is subjective, but knowing the state of the art of CTM modelling, a correlation of 0.5 is considered as "very good" for some species (such as inorganics or PM, for example). Thus, this is important to put this subjective information on a plot to show that the results are not perfect, but may appear as good, knowing the current capabilities of CTMs.

Finally, please note that our answers are in blue in the text and after each reviewers remark.

Best regards,
Laurent MENUT
December 19, 2016

**Message from the Editor**

**Answers to Anonymous Referee #1**

In this paper, the authors present an extension of the evaluation of (atmospheric chemistry) models by using measurements from other years than the year which was simulated by the model. New scores are introduced to quantify the ability of the model to capture the day to day variability as opposed to persistent patterns.
**General comments:**

While reading the paper I asked myself the question if the approach presented by the authors has a real added value as compared to a more traditional model evaluation based on bias, RMS and (one type of) correlation, and may be adopted by other groups. In the end I decided that it probably does, for the following reasons:
- The approach proposed quantifies the importance of day-to-day, weather dominated variability versus systematic patterns which are repeated from year to year.
- The approach naturally leads to an overview of the performance for multiple species in one graph (e.g. Fig.5), which is especially also useful (maybe even more useful) for comparisons between different models. This include both trace species as well as meteorological variables. This is a bit similar to the use of Taylor diagrams.
- The approach explicitly exploits both spatial and temporal correlations, which bring complementary information.
- The approach provides new insight into the performance of the WRF-CHIMERE model. Because of this I am in favour of publication. However, to my opinion there are several major and minor points to be addressed before the paper can be considered by GMD. These are listed below:

- Is this approach really new? The authors provide a few interesting references in the paper, but I would like to see a more systematic overview of the model evaluation approaches and techniques/scores adopted in the past (e.g. including several European/American CTM intercomparison exercises) to better understand the added value of the approach proposed.
  We think the approach is really new: we never see before a comparison between a model simulation and data from other years. We made a complete bibliography, improved in this revised version. This is the novelty of the paper: considering that using other years is the way to split results between "climatological" events and sporadic events and, thus, the model's ability to catch sporadic events.
- The formulation is incomplete, and mathematical formulas are not well defined. In particular, the authors should provide the equations for R_s and R_t, and the mathematical formula for the MYV needs more discussion, see my comments below. Also, the authors should motivate why the R_s,t scores are chosen.
  The part with the mathematical formulas was rewritten and is now more complete. More arguments are proposed for the choice of the MYV formulation. We understand the reviewer comments and, clearly, the score as it proposed may be discussed. In fact, we tried several scores before submitting the publication and we found that the proposed one corresponds to the best choice regarding the type of result we want. The choice of the correlations is detailed below. The bibliography added in the introduction showed that the models are usually validated using three scores: correlation, RMSE and bias. For regulatory purposes, the bias and RMSE are important scores. The bias is certainly the most important to catch the annual mean difference between the model and the observations. But this does not reflect the model variability, i.e the ability of the model to reproduce the real physico-chemical variability. The RMSE is strongly influenced by the bias. For these reasons, we focus on the correlations, spatial and temporal, in this study because we are more interested by the processes evaluation.
- The MYV is not really a model score to my opinion, but rather an indicator of how much the score is influenced by day-to-day variability. In particular one can argue that R=1 and D=0 is a good result. Also, I wonder if a formula for MYV is really needed. Showing D and R is maybe enough (see e.g. Fig. 5)? This should be more carefully presented/discussed.

We agree with the concept of indicator in place of score. This was changed accordingly in the whole text. We think a formula for MYV is really needed because this is the only numerically way to link D and R and to propose a unique value to analyze. Showing D and R is a good way, but mainly a graphical way. In addition, we want values for the discussion, and possibly, inter-model comparisons. Is R=1 and D=0 a good score? Not really, because it means that the model is good to reproduce something easy to model (being every years).

**Detailed comments:**

- p4, l13: "they are used as daily averaged in the present study": why this choice to focus on daily averages instead of hourly values? Please motivate.
  There is two reasons for the use of daily averaged measurements and model outputs: (1) as shown in the table 1, some data are hourly and some others are tri-hourly. In fact, even if we are presented as tri-hourly, the precipitation data are correct to use only in a daily way. As we want to have the same score for all measurements, we then chosen to use daily averaged data. Another reason: we want to split the high temporal frequency variability and the systematic patterns. The day-to-day is the best frequency for that. If we had used the hourly measurements, we certainly added a false variability due to "systematic daily" behaviours such as the diurnal cycle for temperature or $NO_x$ emissions.
- p4, table: Provide also the full names of the variables, e.g. "Temperature at 2m above ground" etc.
  The full name of all variables was added in the text.
- p4, last line: replace "same day for another is" by "same day for another year is"
  OK corrected.
- p5, l4: "The correlation is the more appropriate statistical metric for such analysis." Please explain and motivate this statement in detail. This is important for the rest of the paper!
  This point is similar to a reviewer #2 remark and a long discussion is proposed below. The correlations are able to split the relative contributions of systematic weather or sources dominated variability and day-to-day variability. The key point of this study is the study of the model variability and the variability is statistically represented by the correlations. The mean bias (or the normalized bias) is not a score to quantify the variability. And the RMSE is a score containing a part of variability but is mainly driven by the bias. This was added in the revised version.
- p5, l8: "The spatial correlation, noted Rs, is calculated from the temporal mean averaged values of observations and model for each location where observations are available." Please provide a detailed mathematical formula/recipe to be clear. Are observations and model first collocated for individual observations, or are means computed and then compared. Are these means of daily means or means of hourly values? It is important to define precisely how the correlations are computed: the devil is in the details.
  All correlations are calculated using mean daily values. Using these daily values, the spatial correlation is the correlation using all data, for all sites. The formula for the correlation was added in the revised version.
- p5, l3: Also for the temporal correlation: be more precise. Is it based on daily means, hourly values or something else.
  All scores values are estimated using daily averages values. This was added in the text.
- p5, l14: "The longer the atmospheric lifetime of the species, the lower the relevance of temporal correlation" I would dispute this. For long-lived tracers the transport (wind

direction) and location/strength of the sources becomes crucial, directly influencing temporal correlations. I suggest to remove this remark.
We agree, this remark was removed.

- p5, eq.1: Why is there an absolute value introduced. Instead of absolute(s_i-s_N) I would suggest (S_N minus s_i) assuming higher values of "s" (or "s" close to 1) indicate better performance, which is the case for correlations.
There is an absolute value because all values are not always positive: for some variables and some years, you may have a positive correlation for the year N and a negative one for another year. More difficult, in some cases, you may have a better correlation for another year than for the studied year.

- p6, eq.2: Remove the "X" (multiplication) from the formula. This is not needed (in eq.1 there is also no X). Please introduce a one character symbol for the "Multi Year Variability" instead of writing "MYV" in eq 2, which, in mathematical formula's means M times Y times V. "D_s" has not been introduced: is it the same as "D" ?
The formulas were cleaned and the MYV is now noted $I_{mv}$, for "Indicator of Model Variability".

- p6, eq.2: Why this complicated exponential form?? It seems that you ideally would have the MYV to be =1 for (s=1 and D=1), and =0 for (s=0 or D=0). A much simpler form s_MYV = s_N D would do the trick. In fact, eq.2 is not =1 for s_N=D=1. Where does this formula come from? Is there a reference to a paper introducing this form? Also, it would be good if the formula has clear limits, e.g. 0 (very bad) and 1 (very good). This is not the case when D=1.
The exponential form is really complicated? We think this is easy to implement and to use it. The form was chosen to have a non-linear indicator in order to give more weight to the high values and to take into consideration that the scores (correlation, RMSE or bias) may have a different weight that the differences between years. Of course, the modeller may just use the values of the score and the difference (two values), but the indicator is able to provide just one synthetic value for the discussion.

- p7, l2: Where does the number 0.3 come from? It will depend a lot on how the score "s" is defined. The number seems arbitrarily chosen.
Yes, the value was arbitrarily selected and this is explained in the text, page 6 - line 5. This is a tunable parameter and its only role is to provide a weight on the scores and their differences. The user can change this value as a function of the studied problem. In our case, we found that 0.3 is a good proxy to have values representative of the state-of-the-art of chemistry-transport modelling and validation. As we said, this value is not really important and has no impact on the discussion: this is just a way to highlight the good performances (or not) of the model simulations compared to the observations.

- p7, l14: " ... is challenging because several uncertainties ... "
We agree, we corrected in the revised version.

- Table 2: It would be helpful to remind the reader that these are Summer periods (1-5 to 1-9) and that the scores are based on daily mean values. Please also highlight the special situation for 2013 (I would suggest to start with 2013, add a thick line, and continue with 2008 2009 ... Perhaps it can be stressed once more in the caption that observations for 2008-2012 (and 2013) are compared with 2013 model results.
Yes, we agree with that and for the whole paper, the captions were extended and are now more precise. For the order of the lines, we prefer to keep the increasing order for the years. But the new caption will help to well understand this Table.

- Figure 4: Caption is incomplete.
The caption was completely rewritten and is now more clear.

- Table 3: "... Values of MYV above 0.3 are shown in bold... "
  OK this was corrected.
- p11, l18: ... with differences above 0.5...
  OK this was corrected.

**Answers to Anonymous Referee #2**

This work addresses the important issue of the validation of chemistry transport models. The authors present a new methodology in which the traditional approach consisting of comparing measurements with model results for a given time period is extended to comparisons of the same model results with measurements from other years. The authors develop then a specific indicator on this basis that allows discriminating results that are good for the good reason from those that are good only because of highly persistent pattern present in the observations from year to year. While the proposed methodology is original and has a potential to complement the traditional approach, the authors remain unfortunately superficial and qualitative in their way of presenting and applying this methodology. As a consequence, the proposed examples are qualitative as well and are not helpful. Finally, the document is poorly written: (1) English would need revisions throughout the whole document and (2) many sections would need to be re-written (some suggestions are proposed below).
We thank the reviewer for the interesting suggestions in this review. The English was completely revised and the proposed sections were rewritten.

**Major points:**

1) The authors mention Solazzo and Galmarini (2016) for their decomposition of the error but they finally focus on the correlation only. As noted by these two Authors but also by many others (the referencing to other works relating to model evaluation should be improved), it is important to look at all three possible source of errors because focusing on the only correlation may lead to the wrong conclusions (see comments below). I'm wondering why the Authors make this choice as the proposed methodology could easily be developed for other indicators that are more representative of the overall model performance (e.g. MSE).
This remark is an important and interesting point. Why the scores are done for the correlations (spatial and temporal) and not for the RMSE and the bias? In fact, we did this work in a preliminary version of the paper. Finally, after discussion between all authors, we decided to present only scores for the correlations. We understand this choice may appear surprising but there are several reasons for that:

1. The main goal of this paper is to separate the contributions due to systematic events (i.e the model seems good but finally is only able to model the same thing every day and every year) and due to sporadic events (i.e the model is good because able to retrieve day to day variability). For this goal, the correlations (spatial and temporal) are the most interesting indicators. We agree that RMSE and bias are also important indicators but **the goal of this study is not to replace already existing approaches but to give a complementary insight on the results**.

2. The behaviour of correlations and bias and RMSE is not the same. The correlations are always between 0 and 1. More the correlation is high more the indicator is high. This is the contrary for RMSE and bias: More the score is high more the indicator is low (a large bias indicates a wrong simulation). In addition, the RMSE and bias are

not bounded between 0 and 1, may have large values or negative values. Thus, in a previous version of this paper, we tried to combine the formulation of the indicator with only one formula as:

$$MYV_s = (\alpha - \beta s_N) \times (1 - exp(-D_s)^\delta) \qquad (1)$$

where $\alpha$ and $\beta$ are arbitrarily chosen constants, to define differently depending on the score (correlation or based on absolute values), as:

- For the correlations (Rs and Rt), we want an indicator increasing when the correlations increase. We thus select $\alpha$=0, $\beta$=-1.
- For the bias and RMSE, we want an indicator increasing when the values decrease. We also want that the score is only between 0 and 1 for readability. But, RMSE and bias may be very large. We thus use $\alpha$=1, $\beta$=1 and we impose to have $MYV_s$=0 when negative values are estimated.

The value for $\delta$ is arbitrary but has just to be larger than 1. This tuning parameter enables to adapt the relative weight we want between the absolute value of the scores for the studies year and the differences between all years. In general, we want that a good score for the studied year have a largest weight that the differences: in this case, we select $\delta$=4. By adding RMSE and bias, we are **obliged to have a more complicated formula, with more tuning parameters**.

3. Last: when using these scores with the data presented in the paper, we found no benefit when using RMSE and bias for the discussion. For this letter, we add some results previously found (but not submitted in the paper). This is to show to the reviewer that the use of RMSE and bias is, with this specific approach, not a real benefit for the interpretation of the results. Examples are proposed in Figure 1 for 2m temperature, AOD and O3. But the conclusion is the same for all studied variables: **there is no variability for the RMSE and bias able to help to conclude on the model quality**. A new paragraph is now added in the manuscript to explain this point and why we decided to focus on correlations only.

[Figure]

Figure 1: *Multi years scores for the 2m temperature, ozone and AOD. The reference year is 2013.*

2) The approach proposed by the Authors remains qualitative and the interpretation depend on the setting of an arbitrary threshold (e.g. MYV=0.3 in Figure 3). Throughout the text, the Authors make qualitative judgements (0.6 is good, 0.5 is poor...). This limits the usefulness of the proposed methodology as we never know what a good value of the

indicator is. I do not understand this limitation as it would seem relatively straightforward to calculate a value of the MYV indicator in a similar way but on the only basis of measurements. This observation-based MYV value could then serve as the threshold beyond which model results would be considered good enough.

There is two different things in the paper: (1) the idea to compare a simulation for a specific year with data from another year. This is not qualitative but fully quantitative. (2) the proposal for an indicator, linking the differences between the years and the correlation values, in order to have only one indicator (and not two). This may appear as qualitative because we prefer to say that the user may change this value. But, in reality, we tried a large range of values and we conclude that the proposed value is the best for the problem related to regional chemistry-transport modelling. We changed the text to be clearer: "using $\delta=4$, we consider that the relative weight of the correlation value against the difference reflects well the state-of-the-art of CTMs regional modelling. Using this value, we consider that the model is good enough and for the well reason if MYV>0.3". In addition, even if this seems a good idea, this is not straightforward to establish an "universal" value of the parameters using only observations. Observations are the reality and to compare several years can not provide the information we need. **But, the important thing is that the choice of $\delta$ and MYV>0.3 is not the key point of the paper**. The key point is to use other years that the modelled year to validate the model results. **Please consider these parameters only as an additional help to synthesize and interpret the results**.

3) The document is poorly written. Many sections are unclear and lack sufficient details to be understood. Some suggestions are provided below but the whole document should be thoroughly revised.

Ok, thanks. We made all proposed changes. We are happy to see all these corrections showing the reviewer considers the work is interesting to publish. Detailed answers are provided after each reviewer remark.

**Minor points:**

1. P1, l1: The title is not very representative of the work

   The "unusual way" is the fact that the validation is done using years different from the studied one. To our knowledge (and after an improved bibliography), this is new and unusual.

2. P1, l3: "and by natural" $\rightarrow$ "and natural"

   OK corrected.

3. P1, l19: the transport

   OK corrected.

4. P1, l20: or from the QAERONET

   OK corrected.

5. P2, l1: can be

   OK corrected.

6. P2, l2-3: sentence to be revised

The sentence was too long and was simplified. This is now: *But there can be multiple reasons for a model simulation to agree or disagree with observations. That is because the result of a simulation is the integrated budget of several processes.*

7. P2, l4: "spatial representativeness" → "spatial representativeness of the monitoring stations". In addition, this concept is mentioned for the first time and should be defined. Finally, I do not get the added value of mentioning this here.

   The term is now better defined in the new paragraph (see answer just below for P2L5).

8. P2, l5: "to isolate problems intrinsic to the models,". This is unclear and should be re-phrased

   We agree and the sentence was rewritten and is now more clear as: *A fundamental difference between observations data and models results is the coherence of the spatial representativeness of the monitoring stations compared to the model cell [Valari and Menut, 2008, Solazzo and Galmarini, 2015]. To quantify the model errors due to mis-representation of physics and chemistry from those only due to representativeness, several methodologies have been developed. These methods are effective but often required important computation time.*

9. P2, l6: "relevant": which ones?

   This word was removed in the new version.

10. P2, l7: "but often with huge" → "but often require important"

    OK corrected.

11. P2, l8: references should be within brackets

    OK corrected.

12. P2, l15-17 and l18-20: if the authors cite these works, they should explain in a little bit more detail their main aspects and why these are important in the context of their work. All these references are introduced independently from the scope of the work. For example on l18, what is the decomposition about? L17, what did Rea et al. find that is relevant for this work...

    This part was completely rewritten and new references were added. The work of Real et al. is just cited to show that some studies are dedicated to split the individual contributions. Of course, this is not the same goal as this paper. The reference was removed.

13. P2, l18: scores is often misused in the text. Sometimes as real score, some times meant as correlation. I guess the authors here refer to indicators.

    We agree with this remark and the words "score", "correlation" and "indicator" were harmonized in the paper.

14. P2, l23: "we apply these scores to a model simulation" is unclear. I do not understand how to apply a score to a model simulation. Please check all occurrences of "scores" and check relevance.

    This paragraph was also rewritten. This is now: *For all these variables, temporal and spatial correlations are computed to identify the model capacity compared to observations. First, the correlations are calculated between observations data and model*

*outputs for the simulation year (i.e. the reference year). Second, the correlations are calculated between the observations data for other years and the model output for the reference year. Logically, the correlations calculated for the reference year for observations and model outputs would give the better results. By difference with the correlations calculated for other years (with the observations only), we expect to conclude if the model is able to catch the observed variability and for the good reasons. Using this approach, the goal is to give complementary information to those usually obtained when using only scores (correlations, bias, RMSE) calculated for a single year, the studied year. It is thus expected to give additional elements to answer these questions: Are the performances of the model satisfactory because the model is accurate or just because the model is able to reproduce a situation which is recurrent from year to year? For a given variable, does the model have a good spatial representativeness compared to the corresponding observations?, and Are the biases introduced by meteorological or emissions variability or by the formulation of processes in the chemistry-transport model itself?*

15. P2, l27: provide

    OK corrected (rewritten in the new paragraph).

16. P2, l29: spatial representativeness is not yet defined. Is special representativeness really assessed by this method? I do not believe so (see following comments)

    This is now done with the new paragraph (see answer for P2L5).

17. P2, l33: Score meant as indicator?

    Yes, and it was corrected.

18. P3, figure 1: I do not believe this figure helps understanding. The proposed methodology is quite universal and does not require to enter these details

    This figure is very simple and is just here to illustrate the paragraph. This could be important for people not familiar with the impact of some variables errors on other variables in the chemistry-transport modelling system. But if the reviewer considers this is not useful and this can be a limitation for the publication, we accept to remove this figure.

19. P3, l7: forcings

    The paragraph was completely rewritten.

20. P3, l9-23: these lines are not necessary to the methodology and application

    These lines are not necessary for the methodology application, this is correct. But the knowledge of the several dependencies between the variables helps to the interpretation of the results.

21. P4, l4: unclear

    This was rewritten.

22. P4, l9: for → in

    OK corrected.

23. P4, l12: variable (Table 1)

    OK corrected.

24. P4, l16: and during → for

    OK corrected.

25. P4, l21: take the same day for another → to re-phrase

    Yes, OK. In fact this is "the same date".

26. P5, l4: why is correlation the more appropriate metric. Why couldn't we say the same for the bias, for example?

    Yes, we understand this remark. The reasons for the use of correlation or bias were explained before in this letter. This line was changed as the complete paragraph was rewritten.

27. P5, l5: What is a usual correlation score? A correlation is a correlation and a score a score!

    There is several types of correlations. We added the definition of the Pearson correlation we used in this study.

28. P5, l11-12: I disagree with the authors. A good correlation score does not indicate that the resolution is adequate, transport is adequate... Correlation could be 1 while keeping a huge bias due to a too coarse resolution.

    The reviewer is right if we are talking about absolute value of the variable. But in our case, as indicated P5L9, we are here talking about the location of pollutants plumes (and not their intensity). Our sentence was dedicated to the day to day variability, independently of the bias value.

29. P5, l16: "particularly": why?

    Yes, this is right, there is no reason. This word was deleted.

30. P5, l20: which differences? Between what?

    The differences between the correlations values. The sentence was corrected. But we are here in the paragraph dedicated to the definition of D.

31. P6, l5: why should it be larger than unity?

    Because, at the end, you want to have an indicator between 0 and 1.

32. P6, l5-6: These lines are totally unclear and should be re-phrased

    Yes, OK. This is probably because these lines are unclear that the reviewer was so critical with the principle of an indicator. The paragraph was thus rewritten.

33. P6, l7: have → has

    Ok, the paragraph was completely rewritten.

34. P6, l7: why do we want that a good score... ": although it may appear straightforward, please give a few words of explanation.

    Ok, the paragraph was completely rewritten.

35. P6, l9: What is an academic value of the score, what is the score meaning here?

    The "academic" value is just because the plot does not contain real data but only the values of the indicator. This was added in the text. And we are OK with the wording; this is not "score" here but "indicator".

36. P6, l10: absolute score but also variable: unclear

    OK this was corrected. The text is now: *Ideally we would hope that the model performs well for the correlation scores but also be able to reproduce the observed variability.*

37. P6, l9-15: this all paragraph is unclear and should be rewritten

    This was rewritten.

38. P6, l18-19: 5 times scores in these sentences!

    This was also rewritten.

39. Figure 3 and Figure 6 seems to be inconsistent in terms of X axis labeling.

    There is "correlation" and "score". We replaced "correlation" by "score" in fig 3 for consistency.

40. P7, l1: from Figure 3

    Ok, corrected.

41. P7, l1: we can consider that

    Ok, corrected.

42. P7, l1-2: This means that all conclusions will remain subjective because of this arbitrarily fixed delta parameter. I believe that a measurement based threshold value for delta can be fixed, withdrawing this arbitrary aspect (see major comment above).

    As discussed before, this is not really subjective: the correlations values and the differences values are completely objective. The way to link these two values using the $I_v$ may appear as subjective (because we are fixing a $\delta$ value, but the reviewer has to consider that this is our choice to define an indicator as we want. For the second point, we don't know how to do the same job for observations: the indicator is defined to characterize the model ability to simulate real observed events. The observations alone have not the same meaning: what can we conclude if an observations for the 12 May 2013 is different or not that the same observations for the 12 May of 2008, 2009, 2010... etc? This is not the goal of this paper.

43. P7, l6: done → calculated

    OK corrected.

44. P7, l6: MYV scores

    This was replaced by the new name of the indicator: *To better understand the relevance of $I_v$, two examples are detailed in this section.*

45. P7, l12: vary a lot → vary significantly

    This is P7L13 and this was corrected.

46. P7, l13: is challenging because

    This is P7L14 and this was corrected.

47. P7, l13: again spatial representativeness needs to be defined

    This is now defined in the new paragraph in a previous section.

48. P7, l17: "The spatial correlation is good for all years". I do not understand which arguments the Authors use to state that the score is good. If the spatial pattern is easy to reproduce, it could well be that a correlation of 0.7 should be considered as bad. This seems to be confirmed by the next sentence: "the model reproduces fairly well a spatial patter observed every year". One way forward is to calculate the correlations on the only basis of measurements to get some indicative threshold of what is good or not.

    This remark is close to previous remarks and we rewritten several paragraphs to make it clearer.

49. P8, l2: Are we sure this is for the good reasons?

    If the correlation and the differences are high, we can conclude this is for the good reasons, i.e a correct modelling of the day-to-day variability. In general, the temperature is one of the variables the most well modelled. The result is not surprising.

50. P8, l6: "This species is secondary" seems to contradict p7, l12.

    $NO_2$ is both a primary and a secondary species. This was corrected here.

51. P8, l6,7: I do not agree that a good score for correlation is indicating a good transport, photochemistry... Correlation is indeed only one of the indicators to assess model performances and it only provides a partial vision of model performances. Correlation could be perfect even with a very large bias.

    We agree with that, but here we focus on the emissions and transport in the text. And the correlation is a good indicator for that. The bias is related to the intensity of the source and not to its location or to the transport.

52. P8, l8: low → coarse

    OK corrected.

53. P8, l8: less good → worse

    OK corrected.

54. "Its spatial extent of its representativeness": totally unclear, this should be rephrased

    OK, this was corrected with: ...being more spatially limited (emissions...

55. P8, l18: "The scores": The correlations are calculated, not the scores which are the correlation values

    OK, this was corrected.

56. P8, l20: "each score type". I do not understand what the Authors mean.

    OK. The part "each score type" has no interest since we already defined $I_v$. This was removed.

57. P8, l20: "Results are presented in Table 3. These results... " → Results (Table 3) are discussed...

    OK corrected

58. P8, l24: why only?

    Yes, Ok not "only".

59. P8, l24: Which arguments are used to state that the spatial correlation is not correct? Because the value in the Table is $R_s$=0.09. This was added in the text.

60. P8, l24: for one year → from one year

    OK corrected.

61. P8, l26, 27 and 28: "very good spatial", "less good", "well retrieved". The Authors should explain how they come to these statements.

    We followed the criteria we defined to help the interpretation. Now that the paragraph about the indicator definition is clearer, we think that this part would be also clearer.

62. P8, l31: A few words to explain what the AOD and ANG are would be helpful

    Also following the Reviewer #1, the acronyms were extended. We already removed the figure explaining how a CTM works because the reviewer considers this is too simple and there is no need to remind this in this paper. This is probably the same for the aerosol optical properties, the basis for anyone studying aerosols.

63. Figure 4 caption: Should include explanations of the two curves represented

    Yes, that's right, more informations are added in the caption.

64. P10, l9,10,11: Again I do not agree with these conclusions which cannot be drawn from the only correlation values. Please see all our answers in this letter about the use of the correlations.

65. P11, l19-20: this sentence is unclear

[revised manuscript text omitted]

[Vautard et al., 2012] Vautard, R., Moran, M., Solazzo, E., Gilliam, R., Matthias, V., Bianconi, R., Chemel, C., Ferreira, J., Geyer, B., Hansen, A., Jericevic, A., Prank, M., Segers, A., Silver, J., Werhahn, J., Wolke, R., Rao, S., and Galmarini, S. (2012). Evaluation of the meteorological forcing used for the Air Quality Model Evaluation International Initiative (AQMEII) air quality simulations. *Atmospheric Environment*, 53:15–37.

---

## Author Response (AR2)

**Review of paper gmd-2016-153**
**An unusual way to validate regional chemistry-transport models, L. Menut et al.**

Dear Editor and reviewers,

We acknowledge the reviewers for the time spent to evaluate our work. We also acknowledge the Editor and we made all proposed changes in the revised manuscript.

This is the second revised version for this manuscript. As many points were already discussed, we report the first set of answers to the reviewers at the end of this document.
All our answers are detailed in this letter, but they can be summarized as:

- The English was checked and improved. Some parts of text were moved, shortened or deleted.
- As requested, we added the values of nRMSE for the discussion.
- The fact that the proposed methodology is focused on the model variability (and not on the model errors) is better explained in the text
- The introduction was simplified and some parts are now in the "methodology" section.
- The title was changed and is now: "An alternative way to evaluate chemistry-transport models variability". The word "variability" is explicitely in the title to show we are working more on correlation than on RMSE.
- The principle of the $I_{mv}$ indivator is better explained: this is an indicator corresponding to a weight on the correlation.
- The presentation of the results was improved: the section 4 presents the time series of the correlations and nRMSE for each studied variables, when the section 5 summarizes the results with the Table containing the $I_{mv}$ values and the colored figures representing all the $I_{mv}$ values on a single plot.

Finally, please note that our answers are in blue in the text and after each reviewers remark.

Best regards,
Laurent MENUT
February 18, 2017

**1  Message from the Editor**

Topical Editor Decision:
Reconsider after major revisions (12 Feb 2017) by Slimane Bekki
**Comments to the Author:**
Anonymous Referee #2 has provided a number of corrections and specific suggestions that the authors should take into account.

**2  Second revised version: Answers to Anonymous Referee #1**

**2.1  Major comments**

The methodology proposed by the authors is original and has potential to complement the traditional approach. Unfortunately, as presented in this publication, the added value of this work remains limited because subjective judgements must be made to interpret the results. I also feel that my previous comments have been accounted for, only very partially.

*Answer:*
We thank the reviewer to consider the study as original. We regret his feeling about his comments.

In our first answer (in section **??**, p.**??** of this document), we wrote detailed pages to answer all his points. We consider that a review, and more in a journal called "discussion", is an exchange of ideas. Here, we have the feeling we have to write answers about the same questions. It probably means that our explanations were not enough clear. So, we answer again, trying to be more clear.

1. The bibliography has been extended but those references are not used much in the text. For example the decomposition of the main indicator RMSE (or MSE) into its three components (Solazzo and Galmarini, Thunis et al., Taylor et al.) could be used as starting point to identify the different indicators and justify the choice of the correlation as central one for this study. The decomposition into a systematic and unsystematic error has already been done in other works that are not referenced.

   *Answer:*
   The main goal of the introduction is to present previous works done in the field of "model validation" in a general manner. But it is not to describe in details all previous results. It is to present the general choices of these studies and to clearly show that we are doing differently and explain why. The authors cited by the reviewer have an impressive background in regional CTM validation. They develop tools widely used in the community. In several articles, they showed that they consider that the MSE is the best indicator for model validation. We fully understand this point of view but this is not the goal of our study: we want to focus on correlation only because we think this indicator is sufficient for our observations to model comparisons in the framework of the **model variability**. Our choice is better explained and justified. For the references, there is certainly more to add. The reviewer wrote that "the decomposition... has already been done" but doesn't propose reference. If the reviewer thinks to specific works, why did he not cite them?

2. Even if not applicable in a diagram, information on the bias or on the standard deviation could be provided. Values of SN and D could be calculated and added to the analysis. The fact that the bias is low and does not show any variability (e.g. for T2m) is an interesting information per se. It could also be that the bias shows more variability for other species and then become the crucial parameter to analyse.

   *Answer:*
   As previously explained, the RMSE could be used to calculate differences between years. We added the values in this revised version for the discussion. But as already explained in the previous revision, their addition in the $I_{mv}$ indicator needs to completely change the calculation. We already tried this and this did not provide results having a clear added value compared to the indicators based on the correlations. We added a few explanations about this in the text.

3. My point about the use of the 'score' terminology has not been addressed. This term is used (55 times in the whole document) indifferently to describe the indicator (e.g. p2 l5-6, p3 l63, p3 l87) and the value taken by this indicator (p4 l55), which makes it confusing to understand. Could the Author give a definition of "score" and check that it is consistent though the document?

   *Answer:*
   We agree with this point and this is now corrected.

4. The title does not reflect what the methodology is about. I agree with the Authors that the methodology is unusual but many things can be defined as unusual. I believe that the title should

provide insight on the novel aspects discussed in the document (the use of different meteorological years)

*Answer:*
This is the second time that the reviewer wants to change the title. For this second revised version, we propose: **"An alternative way to evaluate chemistry-transport models variability"** or "Evaluation of model variability using measurements data from other years than the modeled one".

5. Regarding the qualitative aspects of the methodology. I agree that the indicators are calculated quantitatively but the judgement made on whether the results are good or bad remains subjective. This judgement is based on expert knowledge (e.g. that a correlation of 0.5 is very good for a given species). In my view, this limits the benefit of the methodology, as users need to know a-priori what a good behaviour is. I understand that the key point is in the use of several years of data but if at the end the interpretation of the indicator depends on expert judgement, this is a limitation. Examples (p5 l25-27; p5 l47-50; p6 l11)

*Answer:*
This point is about the indicators in a general way. For the correlations, for example, yes, the fact to have a correlation of 0.5 would be good or not, depending on the modelled variables. We agree that the user has to know that a correlation of 0.8 for temperature is the state-of-the-art, when this is closest to 0.5 for $PM_{10}$. The values are different and probably will evolve (and increase) with time and model improvements. In fact, our methodology has to be used by researchers already knowing the atmospheric composition modeling, as well as the current performances for regional air quality modeling. For the $I_{mv}$ indicator, this is the same. It evolves between 0 and 1, considering this indicator is a weighted value of the correlation usually used.

Thus, from our point of view, there is no problem of 'qualitative' discussion in our paper. If the reader is able to know if a correlation is correct or not, he is also able to know if $I_{mv}$ is correct or not. Mainly because the $I_{mv}$ indicator is just a weigthed value of the correlation.

6. I still do not understand why observations cannot be used to fix a minimum threshold. According to me, values of SN and D calculated on the only basis of the set of observations (substituting the model value by the observation of the reference year) could be calculated to make the approach a little bit less subjective.

*Answer:*

We already developed an answer about this point. But it was probably not clear enough.

The $I_{mv}$ indicator is dedicated to the model variability: this is why this is based on the correlations only. This is already written in p3. l66-69 (where the reviewer found this is unclear and this is corrected in this revised version). The main goal of this indicator is to have only one value: the correlation and a weight depending on the other years. This is the originality of the study.

Thus, this indicator has exactly the same meaning than the statiscal indicator used. If we use the temporal correlation, the $I_{mv}$ value is just the result of this indicator multiplied by a weight function (between 0 and 1) and relative to the differences between the years. For example, if you have $R_t$=0.5 for aerosol, you can consider at the regional scale that your model gives satisfactorily results. But, if all years are modelled in the same way, the differences are low, the indicator will automatically decrease.

The use of observations is dedicated to know if the studied year is very different from the others ot not. This is more a problem of trends along years and this has no link with the model quality. In fact the observations are already contained in our calculations: we use the correlation, i.e the way the model is able to reproduce the observed variability. If the difference $D$ is low for one year, it means that the observations evolve in the same way for this another year than for the reference year. To make directly the differences between the observations would give exactly the same result and does not provide an additional 'obsjective' information.

7. English has been improved but many misspells and unclear sentences remain (only few examples provided below).

   *Answer:*
   OK. A second round was done for English.

**2.2 Minor comments**

8. P2 l32 require

9. P2 l64 A couple of lines would be needed to indicate that the Authors now start the description of the methodological approach.

   *Answer:*
   The introduction was simplified and this text was putted in another section, more dedicated to the methodology.

10. P3 l66-69: unclear

11. P3 l72: The bias is an indicator, not a score!

    *Answer:*
    All sentences with score/indicator were checked and corrected.

12. P3 l73: I disagree with the Authors, the RMSE is not driven by bias. Depending on the variable and the period of time considered, the RMSE can be dominated by correlation, bias or by standard deviation.

    *Answer:*
    This sentence was deleted.

13. P3 l78: unclear formulation, please re-phrase.

    *Answer:*
    A large amount of the article was rewritten, including this part.

14. Figure 1: I guess MYV should be Imv

    *Answer:*
    Following this reviewer recommendation, we changed the MYV term by $I_{mv}$ in the first revised version. But, sorry for that, we forgot to change also the first two Figures. This is now corrected.

15. p5 l 23: have → has

16. p5: why a subscript "s" in Ds

    *Answer:*
    Sorry, it was an error and this is now corrected.

17. Figure 2; MYV should become Imv I guess  *Answer:*
    Yes, right and corrected.

18. P5 l70: unclear, please re-phrase

19. P6 l14: year → years

20. P6 l16-18: please re-phrase

21. P6 l27-28: disagree: see point 12

**3 Second revised version: Answers to Anonymous Referee #2**

This reviewer proposed to accept the paper as is.

*Answer:*
We thank this reviewer to have accepted our corrections.

**4 First revised version: Answers to Editor**

There is some common remarks which can be synthesized:

1. *The bibliography could be improved*: this was done and the state of the art regarding the current ways to validate CTMs was rewritten. In brief, the following references were added: [?], [?], [?] [?], [?], [?], [?], [?], [?], [?], [?].

2. *The scores could use RMSE and bias*: This is right, and in fact, we did it during the preparation of the manuscript. This was removed for the submission because we considered that the added-value was low. A long explanation for this choice is proposed in the answer to the reviewer #2.

3. *The interest to have a MYV score*: There is two kinds of novelties in this paper. First, the fact to use data from other years than the studied year is the most important novelty. This is why the title is "unusual way", because this is the first time that such way to estimate the model realism is used. Second, the MYV score. This is also new and the goal is to have a quantified link between the "differences" and the scores (correlation, RMSE, etc.). The constant value is arbitrary, this is true. But the user can select another value. In the case of CTM, this is subjective, but knowing the state of the art of CTM modelling, a correlation of 0.5 is considered as "very good" for some species (such as inorganics or PM, for example). Thus, this is important to put this subjective information on a plot to show that the results are not perfect, but may appear as good, knowing the current capabilities of CTMs.

**5 First revised version: Answers to Anonymous Referee #1**

In this paper, the authors present an extension of the evaluation of (atmospheric chemistry) models by using measurements from other years than the year which was simulated by the model. New scores are introduced to quantify the ability of the model to capture the day to day variability as opposed to persistent patterns.

**General comments:**

While reading the paper I asked myself the question if the approach presented by the authors has a real added value as compared to a more traditional model evaluation based on bias, RMS and (one type of) correlation, and may be adopted by other groups. In the end I decided that it probably does, for the following reasons:

- The approach proposed quantifies the importance of day-to-day, weather dominated variability versus systematic patterns which are repeated from year to year.

- The approach naturally leads to an overview of the performance for multiple species in one graph (e.g. Fig.5), which is especially also useful (maybe even more useful) for comparisons between different models. This include both trace species as well as meteorological variables. This is a bit similar to the use of Taylor diagrams.

- The approach explicitly exploits both spatial and temporal correlations, which bring complementary information.

- The approach provides new insight into the performance of the WRF-CHIMERE model.

Because of this I am in favour of publication. However, to my opinion there are several major and minor points to be addressed before the paper can be considered by GMD.

These are listed below:

- Is this approach really new? The authors provide a few interesting references in the paper, but I would like to see a more systematic overview of the model evaluation approaches and techniques/scores adopted in the past (e.g. including several European/American CTM intercomparison exercises) to better understand the added value of the approach proposed.
  We think the approach is really new: we never see before a comparison between a model simulation and data from other years. We made a complete bibliography, improved in this revised version. This is the novelty of the paper: considering that using other years is the way to split results between "climatological" events and sporadic events and, thus, the model's ability to catch sporadic events.
- The formulation is incomplete, and mathematical formulas are not well defined. In particular, the authors should provide the equations for R_s and R_t, and the mathematical formula for the MYV needs more discussion, see my comments below. Also, the authors should motivate why the R_s,t scores are chosen.
  The part with the mathematical formulas was rewritten and is now more complete. More arguments are proposed for the choice of the MYV formulation. We understand the reviewer comments and, clearly, the score as it proposed may be discussed. In fact, we tried several scores before submitting the publication and we found that the proposed one corresponds to the best choice regarding the type of result we want. The choice of the correlations is detailed below. The bibliography added in the introduction showed that the models are usually validated using three scores: correlation, RMSE and bias. For regulatory purposes, the bias and RMSE are important scores. The bias is certainly the most important to catch the annual mean difference between the model and the observations. But this does not reflect the model variability, i.e the ability of the model to reproduce the real physico-chemical variability. The RMSE is strongly influenced by the bias. For these reasons, we focus on the correlations, spatial and temporal, in this study because we are more interested by the processes evaluation.
- The MYV is not really a model score to my opinion, but rather an indicator of how much the score is influenced by day-to-day variability. In particular one can argue that R=1 and D=0 is a good result. Also, I wonder if a formula for MYV is really needed. Showing D and R is maybe enough (see e.g. Fig. 5)? This should be more carefully presented/discussed.
  We agree with the concept of indicator in place of score. This was changed accordingly in the whole text. We think a formula for MYV is really needed because this is the only numerically way to link D and R and to propose a unique value to analyze. Showing D and R is a good way, but mainly a graphical way. In addition, we want values for the discussion, and possibly, inter-model comparisons. Is R=1 and D=0 a good score? Not really, because it means that the model is good to reproduce something easy to model (being every years).

**Detailed comments:**

- p4, l13: "they are used as daily averaged in the present study": why this choice to focus on daily averages instead of hourly values? Please motivate.
  There is two reasons for the use of daily averaged measurements and model outputs: (1) as shown in the table 1, some data are hourly and some others are tri-hourly. In fact, even if we are presented as tri-hourly, the precipitation data are correct to use only in a daily way. As we want to have the same score for all measurements, we then chosen to use daily averaged data. Another reason: we want to split the high temporal frequency variability and the systematic patterns. The day-to-day is the best frequency for that. If we had used the hourly measurements, we certainly added a false variability due to "systematic daily" behaviours such as the diurnal cycle for temperature or $NO_x$ emissions.
- p4, table: Provide also the full names of the variables, e.g. "Temperature at 2m above ground" etc.
  The full name of all variables was added in the text.

- p4, last line: replace "same day for another is" by "same day for another year is"
  OK corrected.
- p5, l4: "The correlation is the more appropriate statistical metric for such analysis." Please explain and motivate this statement in detail. This is important for the rest of the paper!
  This point is similar to a reviewer #2 remark and a long discussion is proposed below. The correlations are able to split the relative contributions of systematic weather or sources dominated variability and day-to-day variability. The key point of this study is the study of the model variability and the variability is statistically represented by the correlations. The mean bias (or the normalized bias) is not a score to quantify the variability. And the RMSE is a score containing a part of variability but is mainly driven by the bias. This was added in the revised version.
- p5, l8: "The spatial correlation, noted Rs, is calculated from the temporal mean averaged values of observations and model for each location where observations are available." Please provide a detailed mathematical formula/recipe to be clear. Are observations and model first collocated for individual observations, or are means computed and then compared. Are these means of daily means or means of hourly values? It is important to define precisely how the correlations are computed: the devil is in the details.
  All correlations are calculated using mean daily values. Using these daily values, the spatial correlation is the correlation using all data, for all sites. The formula for the correlation was added in the revised version.
- p5, 13: Also for the temporal correlation: be more precise. Is it based on daily means, hourly values or something else.
  All scores values are estimated using daily averages values. This was added in the text.
- p5, l14: "The longer the atmospheric lifetime of the species, the lower the relevance of temporal correlation" I would dispute this. For long-lived tracers the transport (wind direction) and location/strength of the sources becomes crucial, directly influencing temporal correlations. I suggest to remove this remark.
  We agree, this remark was removed.
- p5, eq.1: Why is there an absolute value introduced. Instead of absolute(s_i-s_N) I would suggest (S_N minus s_i) assuming higher values of "s" (or "s" close to 1) indicate better performance, which is the case for correlations.
  There is an absolute value because all values are not always positive: for some variables and some years, you may have a positive correlation for the year N and a negative one for another year. More difficult, in some cases, you may have a better correlation for another year than for the studied year.
- p6, eq.2: Remove the "X" (multiplication) from the formula. This is not needed (in eq.1 there is also no X). Please introduce a one character symbol for the "Multi Year Variability" instead of writing "MYV" in eq 2, which, in mathematical formula's means M times Y times V. "D_s" has not been introduced: is it the same as "D" ?
  The formulas were cleaned and the MYV is now noted $I_{mv}$, for "Indicator of Model Variability".
- p6, eq.2: Why this complicated exponential form?? It seems that you ideally would have the MYV to be =1 for (s=1 and D=1), and =0 for (s=0 or D=0). A much simpler form s_MYV = s_N D would do the trick. In fact, eq.2 is not =1 for s_N=D=1. Where does this formula come from? Is there a reference to a paper introducing this form? Also, it would be good if the formula has clear limits, e.g. 0 (very bad) and 1 (very good). This is not the case when D=1.
  The exponential form is really complicated? We think this is easy to implement and to use it. The form was chosen to have a non-linear indicator in order to give more weight to the high values and to take into consideration that the scores (correlation, RMSE or bias) may have a different weight that the differences between years. Of course, the modeller may just use the values of the score and the difference (two values), but the indicator is able to provide just one synthetic value for the discussion.

- p7, l2: Where does the number 0.3 come from? It will depend a lot on how the score "s" is defined. The number seems arbitrarily chosen.
  Yes, the value was arbitrarily selected and this is explained in the text, page 6 - line 5. This is a tunable parameter and its only role is to provide a weight on the scores and their differences. The user can change this value as a function of the studied problem. In our case, we found that 0.3 is a good proxy to have values representative of the state-of-the-art of chemistry-transport modelling and validation. As we said, this value is not really important and has no impact on the discussion: this is just a way to highlight the good performances (or not) of the model simulations compared to the observations.
- p7, l14: " ... is challenging because several uncertainties ... "
  We agree, we corrected in the revised version.
- Table 2: It would be helpful to remind the reader that these are Summer periods (1-5 to 1-9) and that the scores are based on daily mean values. Please also highlight the special situation for 2013 (I would suggest to start with 2013, add a thick line, and continue with 2008 2009 ... Perhaps it can be stressed once more in the caption that observations for 2008-2012 (and 2013) are compared with 2013 model results.
  Yes, we agree with that and for the whole paper, the captions were extended and are now more precise. For the order of the lines, we prefer to keep the increasing order for the years. But the new caption will help to well understand this Table.
- Figure 4: Caption is incomplete.
  The caption was completely rewritten and is now more clear.
- Table 3: "... Values of MYV above 0.3 are shown in bold... "
  OK this was corrected.
- p11, l18: ... with differences above 0.5...
  OK this was corrected.

**6  First revised version: Answers to Anonymous Referee #2**

This work addresses the important issue of the validation of chemistry transport models. The authors present a new methodology in which the traditional approach consisting of comparing measurements with model results for a given time period is extended to comparisons of the same model results with measurements from other years. The authors develop then a specific indicator on this basis that allows discriminating results that are good for the good reason from those that are good only because of highly persistent pattern present in the observations from year to year. While the proposed methodology is original and has a potential to complement the traditional approach, the authors remain unfortunately superficial and qualitative in their way of presenting and applying this methodology. As a consequence, the proposed examples are qualitative as well and are not helpful. Finally, the document is poorly written: (1) English would need revisions throughout the whole document and (2) many sections would need to be re-written (some suggestions are proposed below).
We thank the reviewer for the interesting suggestions in this review. The English was completely revised and the proposed sections were rewritten.

**Major points:**

1) The authors mention Solazzo and Galmarini (2016) for their decomposition of the error but they finally focus on the correlation only. As noted by these two Authors but also by many others (the referencing to other works relating to model evaluation should be improved), it is important to look at all three possible source of errors because focusing on the only correlation may lead to the wrong conclusions (see comments below). I'm wondering why the Authors make this choice as the proposed

methodology could easily be developed for other indicators that are more representative of the overall model performance (e.g. MSE).

This remark is an important and interesting point. Why the scores are done for the correlations (spatial and temporal) and not for the RMSE and the bias? In fact, we did this work in a preliminary version of the paper. Finally, after discussion between all authors, we decided to present only scores for the correlations. We understand this choice may appear surprising but there are several reasons for that:

1. The main goal of this paper is to separate the contributions due to systematic events (i.e the model seems good but finally is only able to model the same thing every day and every year) and due to sporadic events (i.e the model is good because able to retrieve day to day variability). For this goal, the correlations (spatial and temporal) are the most interesting indicators. We agree that RMSE and bias are also important indicators but **the goal of this study is not to replace already existing approaches but to give a complementary insight on the results**.

2. The behaviour of correlations and bias and RMSE is not the same. The correlations are always between 0 and 1. More the correlation is high more the indicator is high. This is the contrary for RMSE and bias: More the score is high more the indicator is low (a large bias indicates a wrong simulation). In addition, the RMSE and bias are not bounded between 0 and 1, may have large values or negative values. Thus, in a previous version of this paper, we tried to combine the formulation of the indicator with only one formula as:

$$MYV_s = (\alpha - \beta s_N) \times (1 - exp(-D_s)^\delta) \tag{1}$$

where $\alpha$ and $\beta$ are arbitrarily chosen constants, to define differently depending on the score (correlation or based on absolute values), as:

- For the correlations (Rs and Rt), we want an indicator increasing when the correlations increase. We thus select $\alpha$=0, $\beta$=-1.
- For the bias and RMSE, we want an indicator increasing when the values decrease. We also want that the score is only between 0 and 1 for readability. But, RMSE and bias may be very large. We thus use $\alpha$=1, $\beta$=1 and we impose to have $MYV_s$=0 when negative values are estimated.

The value for $\delta$ is arbitrary but has just to be larger than 1. This tuning parameter enables to adapt the relative weight we want between the absolute value of the scores for the studies year and the differences between all years. In general, we want that a good score for the studied year have a largest weight that the differences: in this case, we select $\delta$=4. By adding RMSE and bias, we are **obliged to have a more complicated formula, with more tuning parameters**.

3. Last: when using these scores with the data presented in the paper, we found no benefit when using RMSE and bias for the discussion. For this letter, we add some results previously found (but not submitted in the paper). This is to show to the reviewer that the use of RMSE and bias is, with this specific approach, not a real benefit for the interpretation of the results. Examples are proposed in Figure **??** for 2m temperature, AOD and O3. But the conclusion is the same for all studied variables: **there is no variability for the RMSE and bias able to help to conclude on the model quality**. A new paragraph is now added in the manuscript to explain this point and why we decided to focus on correlations only.

2) The approach proposed by the Authors remains qualitative and the interpretation depend on the setting of an arbitrary threshold (e.g. MYV=0.3 in Figure 3). Throughout the text, the Authors

[Figure]

Figure 1: *Multi years scores for the 2m temperature, ozone and AOD. The reference year is 2013.*

make qualitative judgements (0.6 is good, 0.5 is poor...). This limits the usefulness of the proposed methodology as we never know what a good value of the indicator is. I do not understand this limitation as it would seem relatively straightforward to calculate a value of the MYV indicator in a similar way but on the only basis of measurements. This observation-based MYV value could then serve as the threshold beyond which model results would be considered good enough.

There is two different things in the paper: (1) the idea to compare a simulation for a specific year with data from another year. This is not qualitative but fully quantitative. (2) the proposal for an indicator, linking the differences between the years and the correlation values, in order to have only one indicator (and not two). This may appear as qualitative because we prefer to say that the user may change this value. But, in reality, we tried a large range of values and we conclude that the proposed value is the best for the problem related to regional chemistry-transport modelling. We changed the text to be clearer: "using $\delta$=4, we consider that the relative weight of the correlation value against the difference reflects well the state-of-the-art of CTMs regional modelling. Using this value, we consider that the model is good enough and for the well reason if MYV>0.3". In addition, even if this seems a good idea, this is not straightforward to establish an "universal" value of the parameters using only observations. Observations are the reality and to compare several years can not provide the information we need. **But, the important thing is that the choice of $\delta$ and MYV>0.3 is not the key point of the paper**. The key point is to use other years that the modelled year to validate the model results. **Please consider these parameters only as an additional help to synthesize and interpret the results**.

3) The document is poorly written. Many sections are unclear and lack sufficient details to be understood. Some suggestions are provided below but the whole document should be thoroughly revised.

Ok, thanks. We made all proposed changes. We are happy to see all these corrections showing the reviewer considers the work is interesting to publish. Detailed answers are provided after each reviewer remark.

**Minor points:**

1. P1, l1: The title is not very representative of the work

   The "unusual way" is the fact that the validation is done using years different from the studied one. To our knowledge (and after an improved bibliography), this is new and unusual.

2. P1, l3: "and by natural" $\rightarrow$ "and natural"

   OK corrected.

3. P1, l19: the transport

OK corrected.

4. P1, l20: or from the QAERONET

   OK corrected.

5. P2, l1: can be

   OK corrected.

6. P2, l2-3: sentence to be revised

   The sentence was too long and was simplified. This is now: *But there can be multiple reasons for a model simulation to agree or disagree with observations. That is because the result of a simulation is the integrated budget of several processes.*

7. P2, l4: "spatial representativeness" → "spatial representativeness of the monitoring stations". In addition, this concept is mentioned for the first time and should be defined. Finally, I do not get the added value of mentioning this here.

   The term is now better defined in the new paragraph (see answer just below for P2L5).

8. P2, l5: "to isolate problems intrinsic to the models,". This is unclear and should be re-phrased

   We agree and the sentence was rewritten and is now more clear as: *A fundamental difference between observations data and models results is the coherence of the spatial representativeness of the monitoring stations compared to the model cell [?, ?]. To quantify the model errors due to mis-representation of physics and chemistry from those only due to representativeness, several methodologies have been developed. These methods are effective but often required important computation time.*

9. P2, l6: "relevant": which ones?

   This word was removed in the new version.

10. P2, l7: "but often with huge" → "but often require important"

    OK corrected.

11. P2, l8: references should be within brackets

    OK corrected.

12. P2, l15-17 and l18-20: if the authors cite these works, they should explain in a little bit more detail their main aspects and why these are important in the context of their work. All these references are introduced independently from the scope of the work. For example on l18, what is the decomposition about? L17, what did Rea et al. find that is relevant for this work...

    This part was completely rewritten and new references were added. The work of Real et al. is just cited to show that some studies are dedicated to split the individual contributions. Of course, this is not the same goal as this paper. The reference was removed.

13. P2, l18: scores is often misused in the text. Sometimes as real score, some times meant as correlation. I guess the authors here refer to indicators.

    We agree with this remark and the words "score", "correlation" and "indicator" were harmonized in the paper.

14. P2, l23: "we apply these scores to a model simulation" is unclear. I do not understand how to apply a score to a model simulation. Please check all occurrences of "scores" and check relevance.

    *This paragraph was also rewritten. This is now: For all these variables, temporal and spatial correlations are computed to identify the model capacity compared to observations. First, the correlations are calculated between observations data and model outputs for the simulation year (i.e. the reference year). Second, the correlations are calculated between the observations data for other years and the model output for the reference year. Logically, the correlations calculated for the reference year for observations and model outputs would give the better results. By difference with the correlations calculated for other years (with the observations only), we expect to conclude if the model is able to catch the observed variability and for the good reasons. Using this approach, the goal is to give complementary information to those usually obtained when using only scores (correlations, bias, RMSE) calculated for a single year, the studied year. It is thus expected to give additional elements to answer these questions: Are the performances of the model satisfactory because the model is accurate or just because the model is able to reproduce a situation which is recurrent from year to year? For a given variable, does the model have a good spatial representativeness compared to the corresponding observations?, and Are the biases introduced by meteorological or emissions variability or by the formulation of processes in the chemistry-transport model itself?*

15. P2, l27: provide

    OK corrected (rewritten in the new paragraph).

16. P2, l29: spatial representativeness is not yet defined. Is special representativeness really assessed by this method? I do not believe so (see following comments)

    This is now done with the new paragraph (see answer for P2L5).

17. P2, l33: Score meant as indicator?

    Yes, and it was corrected.

18. P3, figure 1: I do not believe this figure helps understanding. The proposed methodology is quite universal and does not require to enter these details

    This figure is very simple and is just here to illustrate the paragraph. This could be important for people not familiar with the impact of some variables errors on other variables in the chemistry-transport modelling system. But if the reviewer considers this is not useful and this can be a limitation for the publication, we accept to remove this figure.

19. P3, l7: forcings

    The paragraph was completely rewritten.

20. P3, l9-23: these lines are not necessary to the methodology and application

    These lines are not necessary for the methodology application, this is correct. But the knowledge of the several dependencies between the variables helps to the interpretation of the results.

21. P4, l4: unclear

    This was rewritten.

22. P4, l9: for → in

    OK corrected.

23. P4, l12: variable (Table 1)

    OK corrected.

24. P4, l16: and during → for

    OK corrected.

25. P4, l21: take the same day for another → to re-phrase

    Yes, OK. In fact this is "the same date".

26. P5, l4: why is correlation the more appropriate metric. Why couldn't we say the same for the bias, for example?

    Yes, we understand this remark. The reasons for the use of correlation or bias were explained before in this letter. This line was changed as the complete paragraph was rewritten.

27. P5, l5: What is a usual correlation score? A correlation is a correlation and a score a score!

    There is several types of correlations. We added the definition of the Pearson correlation we used in this study.

28. P5, l11-12: I disagree with the authors. A good correlation score does not indicate that the resolution is adequate, transport is adequate... Correlation could be 1 while keeping a huge bias due to a too coarse resolution.

    The reviewer is right if we are talking about absolute value of the variable. But in our case, as indicated P5L9, we are here talking about the location of pollutants plumes (and not their intensity). Our sentence was dedicated to the day to day variability, independently of the bias value.

29. P5, l16: "particularly": why?

    Yes, this is right, there is no reason. This word was deleted.

30. P5, l20: which differences? Between what?

    The differences between the correlations values. The sentence was corrected. But we are here in the paragraph dedicated to the definition of D.

31. P6, l5: why should it be larger than unity?

    Because, at the end, you want to have an indicator between 0 and 1.

32. P6, l5-6: These lines are totally unclear and should be re-phrased

    Yes, OK. This is probably because these lines are unclear that the reviewer was so critical with the principle of an indicator. The paragraph was thus rewritten.

33. P6, l7: have → has

    Ok, the paragraph was completely rewritten.

34. P6, l7: why do we want that a good score... ": although it may appear straightforward, please give a few words of explanation.

    Ok, the paragraph was completely rewritten.

35. P6, l9: What is an academic value of the score, what is the score meaning here?

    The "academic" value is just because the plot does not contain real data but only the values of the indicator. This was added in the text. And we are OK with the wording; this is not "score" here but "indicator".

36. P6, l10: absolute score but also variable: unclear

    OK this was corrected. The text is now: *Ideally we would hope that the model performs well for the correlation scores but also be able to reproduce the observed variability.*

37. P6, l9-15: this all paragraph is unclear and should be rewritten

    This was rewritten.

38. P6, l18-19: 5 times scores in these sentences!

    This was also rewritten.

39. Figure 3 and Figure 6 seems to be inconsistent in terms of X axis labeling.

    There is "correlation" and "score". We replaced "correlation" by "score" in fig 3 for consistency.

40. P7, l1: from Figure 3

    Ok, corrected.

41. P7, l1: we can consider that

    Ok, corrected.

42. P7, l1-2: This means that all conclusions will remain subjective because of this arbitrarily fixed delta parameter. I believe that a measurement based threshold value for delta can be fixed, withdrawing this arbitrary aspect (see major comment above).

    As discussed before, this is not really subjective: the correlations values and the differences values are completely objective. The way to link these two values using the $I_v$ may appear as subjective (because we are fixing a $\delta$ value, but the reviewer has to consider that this is our choice to define an indicator as we want. For the second point, we don't know how to do the same job for observations: the indicator is defined to characterize the model ability to simulate real observed events. The observations alone have not the same meaning: what can we conclude if an observations for the 12 May 2013 is different or not that the same observations for the 12 May of 2008, 2009, 2010... etc? This is not the goal of this paper.

43. P7, l6: done → calculated

    OK corrected.

44. P7, l6: MYV scores

    This was replaced by the new name of the indicator: *To better understand the relevance of $I_v$, two examples are detailed in this section.*

45. P7, l12: vary a lot → vary significantly

    This is P7L13 and this was corrected.

46. P7, l13: is challenging because

    This is P7L14 and this was corrected.

47. P7, l13: again spatial representativeness needs to be defined

This is now defined in the new paragraph in a previous section.

48. P7, l17: "The spatial correlation is good for all years". I do not understand which arguments the Authors use to state that the score is good. If the spatial pattern is easy to reproduce, it could well be that a correlation of 0.7 should be considered as bad. This seems to be confirmed by the next sentence: "the model reproduces fairly well a spatial patter observed every year". One way forward is to calculate the correlations on the only basis of measurements to get some indicative threshold of what is good or not.

This remark is close to previous remarks and we rewritten several paragraphs to make it clearer.

49. P8, l2: Are we sure this is for the good reasons?

If the correlation and the differences are high, we can conclude this is for the good reasons, i.e a correct modelling of the day-to-day variability. In general, the temperature is one of the variables the most well modelled. The result is not surprising.

50. P8, l6: "This species is secondary" seems to contradict p7, l12.

$NO_2$ is both a primary and a secondary species. This was corrected here.

51. P8, l6,7: I do not agree that a good score for correlation is indicating a good transport, photo-chemistry... Correlation is indeed only one of the indicators to assess model performances and it only provides a partial vision of model performances. Correlation could be perfect even with a very large bias.

We agree with that, but here we focus on the emissions and transport in the text. And the correlation is a good indicator for that. The bias is related to the intensity of the source and not to its location or to the transport.

52. P8, l8: low → coarse

OK corrected.

53. P8, l8: less good → worse

OK corrected.

54. "Its spatial extent of its representativeness": totally unclear, this should be rephrased

OK, this was corrected with: ...being more spatially limited (emissions...

55. P8, l18: "The scores": The correlations are calculated, not the scores which are the correlation values

OK, this was corrected.

56. P8, l20: "each score type". I do not understand what the Authors mean.

OK. The part "each score type" has no interest since we already defined $I_v$. This was removed.

57. P8, l20: "Results are presented in Table 3. These results... " → Results (Table 3) are discussed...

OK corrected

58. P8, l24: why only?

Yes, Ok not "only".

59. P8, l24: Which arguments are used to state that the spatial correlation is not correct?

Because the value in the Table is $R_s$=0.09. This was added in the text.

60. P8, l24: for one year → from one year

OK corrected.

61. P8, l26, 27 and 28: "very good spatial", "less good", "well retrieved". The Authors should explain how they come to these statements.

We followed the criteria we defined to help the interpretation. Now that the paragraph about the indicator definition is clearer, we think that this part would be also clearer.

62. P8, l31: A few words to explain what the AOD and ANG are would be helpful

Also following the Reviewer #1, the acronyms were extended. We already removed the figure explaining how a CTM works because the reviewer considers this is too simple and there is no need to remind this in this paper. This is probably the same for the aerosol optical properties, the basis for anyone studying aerosols.

63. Figure 4 caption: Should include explanations of the two curves represented

Yes, that's right, more informations are added in the caption.

64. P10, l9,10,11: Again I do not agree with these conclusions which cannot be drawn from the only correlation values. Please see all our answers in this letter about the use of the correlations.

65. P11, l19-20: this sentence is unclear

Ok, the sentence was changed. This is now: *The low values of correlations show that some variables are systematically badly estimated. This means that some meteorological structures (for $u_{10m}$) or emission sources (contributing to the $PM_{2.5}$ surface concentrations) are systematically mis-located.*

66. P12, l29: dued → due

Oups. OK, thanks, this was corrected.

[revised manuscript text omitted]